# Optimizing power grids: A valley-filling heuristic for energy-efficient electric vehicle charging

**Guilherme Gloriano de Souza** [1☯]*, **Ricardo Ribeiro dos Santos** [1☯], **Ruben Barros Godoy** [2]

**1** College of Computing, Federal University of Mato Grosso do Sul, Campo Grande, Mato Grosso do Sul, Brazil, **2** College of Engineering, Faculty of Engineering, Architecture and Urbanism and Geography, Federal University of Mato Grosso do Sul, Campo Grande, Mato Grosso do Sul, Brazil

☯ These authors contributed equally to this work.

* guilherme.gloriano@ufms.br

**Data Availability Statement:** Souza, G. G. (2024, November 24). Optimizing power grids a VF heuristic. https://doi.org/10.17605/OSF.IO/7R8GZ.

## Abstract

The expansion of electric vehicles (EVs) challenges electricity grids by increasing charging demand, thereby making Demand-Side Management (DSM) strategies essential to maintaining balance between supply and demand. Among these strategies, the Valley-Filling approach has emerged as a promising method to optimize renewable energy utilization and alleviate grid stress. This study introduces a novel heuristic, Load Conservation Valley-Filling (LCVF), which builds on the Classical and Optimistic Valley-Filling approaches by incorporating dynamic load conservation principles, enabling better alignment of EV charging with grid capacity. We conducted a comprehensive analysis of the heuristic across five EV charging scenarios. In both the Original and Flexible scenarios, LCVF reduced energy demand by up to 10.65%, demonstrating its adaptability and effectiveness. Notably, in the 24-hour Availability scenario, LCVF achieved a reduction of over 20% in energy demand compared to CVF. These findings indicate that LCVF could play a crucial role in enhancing real-world EV charging infrastructure, boosting energy efficiency and grid stability. By integrating DSM strategies like LCVF, energy grids can better accommodate renewable energy sources, promoting more sustainable operations.

## 1 Introduction

The rapid adoption of electric vehicles (EVs) has posed significant challenges to existing electricity grids, primarily due to the increased demand for charging. This surge in demand not only strains grid stability but also necessitates structural adjustments to manage this impact, underscoring the urgency of adapting electrical infrastructure to meet the growing EV demand [1]. To address these challenges, Demand Side Management (DSM) strategies emerge as essential tools for efficiently balancing electricity supply and demand [2, 3]. DSM refers to strategies and technologies that encourage consumers to adjust their electricity usage by either reducing consumption during peak demand or shifting it to off-peak times [4]. DSM techniques encompass various methods: peak clipping, valley-filling, load shifting, strategic conservation,

**Funding:** Conselho Nacional de Desenvolvimento Científico e Tecnológico Award Number: 160179/2020-3.

**Competing interests:** NO authors have competing interests.

strategic load growth, and flexible load shaping [5, 6]. Among these, Valley-Filling (VF) has gained attention as a promising method to optimize energy use and alleviate grid stress [7]. Valley-Filling, specifically, aims to increase energy use during periods of low demand (valleys) to balance the load on the electricity grid more efficiently [8]. As EV adoption grows, innovative load allocation strategies are essential to integrate these high-demand elements without risking grid overload. With increasing EV adoption, power demand variability intensifies, particularly during peak hours [9]. This variability places considerable stress on grid infrastructure, potentially leading to power supply failures, increased operational costs, and substantial investments in grid upgrades [10].

The Valley-Filling strategy encourages EV charging during low-demand periods, maximizing the utilization of surplus grid capacity and enhancing stability. Additionally, Valley-Filling improves operational cost efficiency by smoothing energy demand throughout the day, reducing the need to activate supplementary energy sources during peak periods [11–13]. In contrast, other DSM techniques, such as peak-shaving or load-shifting, focus on peak reduction without fully utilizing available capacity and often require additional resources to meet growing power needs. However, a gap exists in the literature regarding the development of heuristics that integrate flexible and adaptive load allocation methods to enhance energy efficiency and grid stability amid increasing EV demand. To address this gap, this study introduces the Load Conservation Valley-Filling (LCVF) heuristic, an innovative approach based on classical Valley-Filling, aimed at optimizing load allocation across diverse consumption scenarios.

This article introduces the Load Conservation Valley-Filling (LCVF) heuristic, a novel approach in EV charging management. The LCVF combines elements from both Classical (CVF) and Optimistic (OVF) valley-filling strategies. It analyzes daily load time-slots (TS) to identify when EV power demands can be allocated to achieve a peak charge (Pc), which represents the maximum charge required to accommodate the EVs. A time slot (TS) refers to a specific interval during which the electricity demand is measured, allowing for detailed control of energy usage over time. The peak charge ($Pc$) represents the maximum power demand observed during these intervals, often requiring careful management to avoid overloading the grid. The OVF approach focuses on valley points in the daily load TS for EV demand allocation, while LCVF builds on this by leveraging charge allocation states of EVs across iterations. This heuristic operates as a daily prediction algorithm, enabling grid operators to analyze previous patterns of EV consumption to determine whether to allocate additional power supply to meet user demands.

This article introduces the Load Conservation Valley-Filling (LCVF) heuristic, a novel approach in EV charging management. The LCVF combines elements from both Classical (CVF) and Optimistic (OVF) valley-filling strategies. It analyzes daily load time-slots (TS) to identify when EV power demands can be allocated to achieve a peak charge (Pc), the maximum required to support EV charging needs. A time slot (TS) refers to a specific interval for measuring electricity demand, enabling detailed control of energy usage over time. The OVF approach targets valley points in the daily load TS for EV demand allocation, while LCVF builds on this by incorporating iterative EV charge allocation states. This heuristic functions as a daily predictive algorithm, empowering grid operators to evaluate past EV consumption patterns and decide on additional power allocations to meet user demand effectively.

We evaluate the proposed heuristic across various EV power demand scenarios, focusing on the maximum power (peak charge) allocated to electric vehicles and each heuristic's performance (execution time). Additionally, we analyze and compare the results of the CVF, OVF, and LCVF approaches. The findings demonstrate that LCVF can improve EV power demand allocation, achieving up to a 20% reduction in peak consumption for recharging electric vehicles while enhancing grid stability. This analysis encourages charging during off-peak hours,

leading to improved grid stability, potential cost reductions for consumers and providers, and a decreased need for immediate grid infrastructure upgrades.

In this context, we found that the main contributions of this paper are:

- Proposal of a new heuristic based on the valley-filling approach for EV power charging allocation.

- A comprehensive and detailed set of experiments using scenarios that could meet different EV users demands.

- Design, organization, and availability of datasets from "classic" user-consumption patterns, which could be beneficial for future research requiring extensive datasrets for testing and validation.

This paper is organized as follows: Section II reviews the literature on Demand Side Management (DSM) and Valley-Filling strategies integrating Electric Vehicles (EVs). Section III presents the Classical and the Optimistic heuristics. Section IV provides the details of Load Conservation Valley-Filling (LCVF). Section V describes the experimental scenarios for evaluating the heuristics. Section VI presents and compares the experimental results of OVF, LCVF, and Classic Valley-Filling (CVF). Finally, Section VII presents the conclusions of the work, highlighting the significant energy efficiency and load management benefits of the proposed algorithm.

## 2 Related work

The Valley-Filling technique is a key aspect of DSM strategies, particularly for optimizing EV charging. These strategies have been extensively studied and widely utilized in academic research [14–16]. This section reviews the literature on DSM approaches in the context of EV integration and grid stability, focusing on Valley-Filling methodologies, EV aggregators, V2G technologies, and forecasting algorithms.

This literature review focuses on DSM approaches that introduce new EV load allocation strategies, highlight experimental results, and propose practical solutions to challenges such as peak demand reduction and grid stability improvement. The study by Abdelfattah et al. [17] proposes a demand response model for fast-charging stations for electric vehicles, integrating user demand with grid capacity. This model stands out for considering both the limitations of the electrical infrastructure and the need for rapid charging by users, making it especially relevant in settings where load variability strains the grid. The results indicate that demand response strategies can reduce operational costs and improve energy efficiency, aligning well with Valley-Filling and DSM approaches designed to stabilize the grid and reduce consumption peaks.

Khan et al. [18] proposed a strategy to coordinate multiple EV aggregators, leveling the load profile and reducing demand peaks. The strategy was tested on the IEEE 13-Node distribution system in Seoul using real-world mobility data and compared with an uncontrolled loading scenario. Key performance indices included reductions of 85% in the peak shaving index, 90% in the valley filling index, and 25% in load variance. These results demonstrate the effectiveness of coordinated EV charging in mitigating grid stress.

Arango et al. [19] proposed a control algorithm for vehicle-to-grid (V2G) technology aimed at reducing load peaks in public infrastructure. The algorithm optimizes the charging/discharging schedule for parked EVs using load profiles and parameters such as the state-of-charge (SoC) and user mobility needs. Tested across three parking scenarios, the algorithm achieved a peak demand reduction of up to 18.2% and an average load factor improvement of

8.7%. These results highlight the potential of V2G to reduce peak loads and enhance grid stability.

Gautam et al. [20] developed a smart grid model featuring adaptive load forecasting and vehicle-to-grid (V2G) integration to address load uncertainty and mitigate grid overloading. The algorithm reduced user electricity costs by implementing Time-of-Use (ToU) pricing, optimizing EV scheduling accordingly. The model was evaluated using load data from Toronto, achieving a Mean Absolute Percentage Error (MAPE) of 4.11%. This study effectively compared the model with traditional forecasting methods, such as ARIMA, demonstrating its superior performance.

Wang et al. [21] used a GA-A* strategy combining a genetic algorithm (GA) with the A* algorithm to enhance EV routing and energy savings. The technique leverages A*'s efficiency in initializing the GA population, which helps avoid local optima and accelerates iteration speed. The GA-A* algorithm outperformed traditional GA, LNS, VNS, and ALNS, achieving a notable reduction in energy consumption. In one benchmark case, GA-A* saved 334.84 kW compared to traditional GA. The average energy savings underscore the robustness of GA-A* in supporting valley-filling applications.

The study by Jian, Zheng, and Shao [22] presents a valley-filling algorithmic strategy for coordinating large-scale EV charging. The algorithm leverages capacity margin and charging priority indices to optimize charging schedules, effectively reducing peak loads. In simulations involving up to 2 million EVs, the algorithm reduced the median peak load from 17, 547 MW to 15, 157 MW. These results underscore the potential of valley-filling strategies in managing large-scale EV charging and stabilizing grid operations.

The studies discussed here explore a variety of DSM approaches, from coordinating EV aggregators [18] and implementing V2G technologies [19–21] to managing large-scale EV charging [22]. These methods emphasize enhancing grid stability, optimizing EV battery use, and reducing operational costs. In this study, the proposed LCVF heuristic builds on these approaches, addressing gaps in computational efficiency and scalability for effectively managing large-scale EV charging.

Table 1 summarizes the key methodologies, findings, and limitations of the studies reviewed, illustrating how the proposed LCVF heuristic addresses specific limitations. While existing studies provide valuable solutions for EV charging, many fail to adequately address oscillatory grid behavior. The proposed LCVF heuristic fills this gap, offering scalability and high performance across different charging scenarios. Evaluations across various EV charging scenarios demonstrate LCVF's potential to reduce oscillations and enhance grid stability, making it a valuable tool for grid operators.

**Table 1. Summary of literature on DSM techniques for EV integration.**

| Author(s) | Methodology | Key Findings | Limitations |
|---|---|---|---|
| Abdelfattah et al. [17] | Demand response model for EV fast-charging | Reduced operational costs and increased efficiency | Tested only in fast-charging stations; lacks large-scale validation |
| Khan et al. [18] | Coordinated charging of multiple EV aggregators to level load profile | Reduced peak demand by 85%, filled valleys by 90%, and reduced load variance by 25% | Focuses on single test system (IEEE 13-Node), lacks generalizability |
| Arango et al. [19] | Vehicle-to-grid (V2G) control algorithm for public infrastructure | Reduced peak demand by 13.5% to 18.2%, improved load factor by 8.7% | Validated with limited parking scenarios, lacks scalability |
| Gautam et al. [20] | Adaptive load forecasting using multilayer perceptron (MLP) model with ToU pricing | MAPE of 4.11%, reduced user electricity costs, accurate daily load predictions | Does not account for real-time grid balancing, only focuses on historical data |
| Wang et al. [21] | GA-A* hybrid algorithm for electric vehicle routing problem with time windows (EVRPTW) | Reduced energy consumption significantly compared to traditional GA and LNS algorithms | Higher computational cost due to integration of GA and A* |
| Jian et al. [22] | Large-scale valley-filling strategy for coordinated EV charging | Median peak load reduction from 17,547 MW to 15,157 MW with 2 million EVs | Limited to simulated data, no real-world validation |

## 3 Valley-filling design

Valley-Filling (VF) is one of the primary Demand Side Management (DSM) approaches to optimize the use of electrical grid resources according to energy demands, aiming to fill periods of lower demand, thus maximizing the efficiency of the electrical system. The VF approach divides the 24-hour daily interval into equivalent time slices, for example, 10-minute intervals. When conventional consumption decreases, this results in a depression in the electrical energy demand curve, known as a valley. Fig 1 depicts an example of a conventional pattern of power user consumption over a day. The power consumption (y-axis) is divided into TS of 10 min (x-axis). The peak is the TS with the highest load demand. A valley corresponds to the TS with the lowest demand recorded in the electricity grid.

In the context of electric vehicles that depend on the electrical grid for recharging, the VF approach seeks to schedule the charging demands to periods of lower conventional energy consumption, that is, in the valleys of the grid demand curve. Using VF to schedule EV charging demands aims to distribute the energy demand over intervals where conventional electrical consumption is lower (valley), reducing the need to recharge them in periods of high demand (peak).

Designing VF-based heuristics with large-scale electric vehicle demands represents a complex challenge for electric vehicle aggregators [23]. The relationship between computational complexity and grid stress is critical in achieving viable solutions. VF-based algorithms have been used to ensure acceptable runtimes when performing for large-scale electric vehicle demands [24].

This section will describe two heuristics based on the valley-filling approach applied to electric vehicle charging balancing. The first heuristic, "Classic Valley-Filling" (CVF), was proposed in [22]. Another heuristic is the "Optimistic Valley-Filling" (OVF) that increases the power range in which the power demands of electric vehicles could be allocated. OVF uses the same allocation strategy as CVF but sets the $Pc$ parameter by looking at the valley of the conventional load (the power consumption pattern of the grid without EVs).

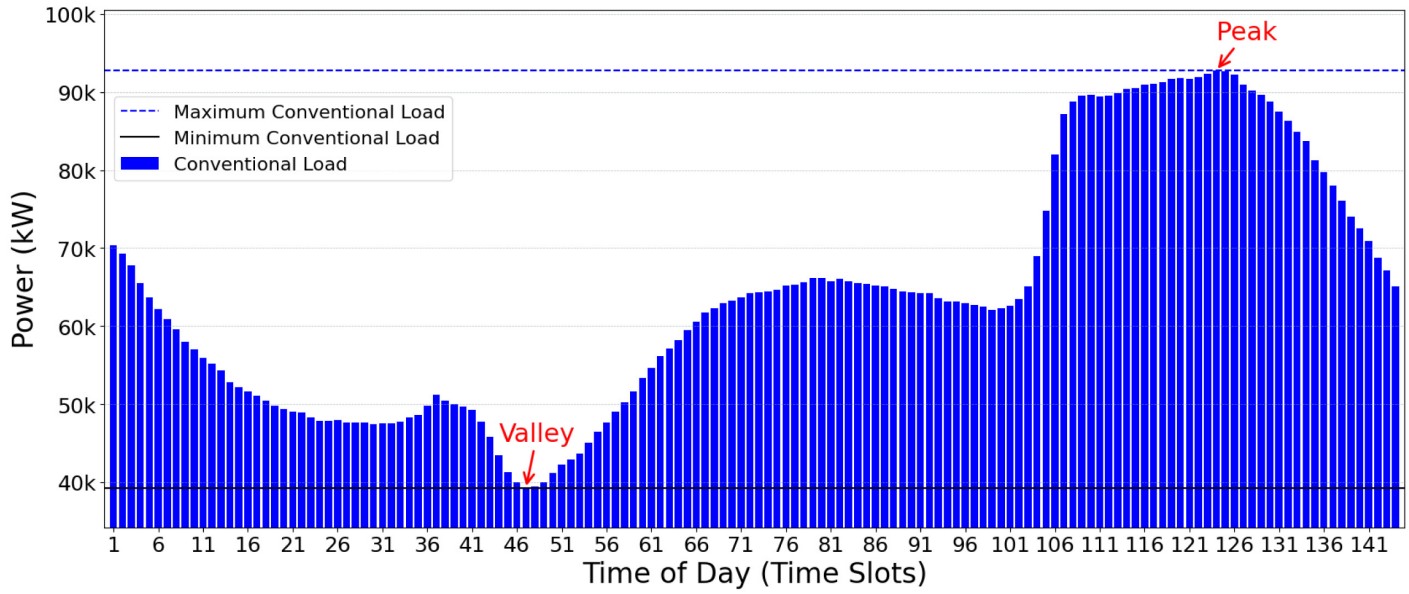

**Fig 1. Example of conventional electricity demand (without data from electric vehicle recharging).**

### 3.1 Classic valley-filling (CVF)

The algorithm presented by Jian et al. [22] employs a greedy strategy to find valleys in the daily grid power consumption profile (Algorithm 1). Once a valley is identified, the algorithm allocates resources up to an upper limit. The algorithm has as inputs the data of the conventional power consumption (*dataEn*), the power charging demands of the EVs (*dataEv*), and the Peak charge (*Pc*) criterion, which is the upper power constraint to each allocation. While there are available time-slots, the algorithm calculates the excess power and the total power demand for each TS (lines 3-4); it determines the capacity margin index for each interval, identifying those with the highest margin (lines 5-6). The capacity margin index measures the available capacity between the current demand level and the maximum allowable power (*Pc*) in each interval, helping the algorithm prioritize slots with greater remaining capacity for allocation. If the margin index is greater than or equal to 1, loads are allocated directly to this range (lines 7-8). Otherwise, a charging priority index is calculated for each EV, vehicles are ranked according to that priority, and charges are allocated based on that (lines 10-12). After each allocation, each EV's charging data is updated (line 14). If all loading demands are met, the algorithm terminates (lines 15-17); otherwise, the gap with the largest margin is removed from the gap set (line 19), and the process repeats until all vehicles are loaded or no more slots are available.

**Algorithm 1** Classic Valley-Filling (CVF)

```
 1: function CVF_SCHEDULING(dataEn, dataEv, Pc)
 2:   while Set of Time-Slots > 0 do
 3:     Calculate surplus power for each TS
 4:     Calculate total demanded power for each TS
 5:     Calculate capacity margin index for each TS
 6:     Select the TS with the highest margin index (valley)
 7:     if margin index ≥ 1 then                          ▷ Case 1
 8:       Allocate charging loads directly
 9:     else                                              ▷ Case 2
10:       Calculate charging priority index for each EV
11:       Sort vehicles according to priority index
12:       Allocate charging loads based on priority
13:     end if
14:     Update charging data for each EV
15:     if All EV charging demands are satisfied then
16:       Flag ← 1
17:       break
18:     end if
19:     Remove the valley from the TS set
20:   end while
21: end function
```

CVF assumes that, at each time interval, the available power from the valleys of the conventional demand curve to its peak is considered surplus and can be allocated to meet the energy demands of electric vehicles. The maximum value of the conventional electrical energy peak is determined by the delimiting parameter (*Pc*), indicating the maximum amount of charge that can be attributed to electric vehicles in a given time interval.

Algorithm 2 optimizes the distribution of electric vehicle (EV) loads using electrical energy demand valleys. It receives as input the available energy data (*dataEn*), EV demand data (*dataEv*), the maximum power level considering electric vehicles (*upperPc*) and the maximum conventional power level without electric vehicles (*lowerPc*) (Eq 1). Initially, *upperPc* is defined as the maximum power level required to accommodate EV loads, while *lowerPc* represents the power level limit without the additional EV load. According to the number of iterations (set by the user), the algorithm iteratively calculates the average Peak charge (*Pc*) criterion and calls

the **cvf_scheduling** function (Algorithm 1). This function is responsible for scheduling and distributing the EV power loads across the electricity consumption valleys. In each iteration, if the function indicates that all loading demands are satisfied (*Flag* = 1), *upperPc* is updated to the current *Pc* value to narrow the search range for an optimal solution; otherwise, *lowerPc* is adjusted to the current *Pc* to increase the power allocation threshold. This bisection method is necessary for converging on the minimum achievable *Pc* and continues until the predefined number of iterations is reached.

$$Pc \quad = \frac{upperPc + lowerPc}{2}$$
$$= \frac{\boldsymbol{max}(EVsLoad + Conv.Load) + \boldsymbol{max}(Conv.Load)}{2} \tag{1}$$

**Algorithm 2** Valley-Filling Charging Scheduling

```
1: function MAIN_VF(dataEn, dataEv, upperPc, lowerPc)
2:    for the chosen number of iterations do
3:        Pc ← (upperPc + lowerPc)/2
4:        Flag ← CVF_scheduling(dataEn, dataEv, Pc)
5:        if Flag equals 1 then
6:           upperPc ← Pc
7:        else
8:           lowerPc ← Pc
9:        end if
10:   end for
11: end function
```

Running this algorithm consists of searching, in each iteration, for the most suitable value of *Pc*, where it is possible to meet the charging demand for electric vehicles. At each iteration, if the power demand of electric vehicles is reached, the *Pc* is decreased, reducing the power supply to meet the demand. Otherwise, *Pc* will have a higher value in the next iteration, indicating the need for a higher power supply (to meet the power demand) for electric vehicles than in the previous iteration.

A possible use-case input scenario for the CVF algorithm is illustrated in Fig 2. The daily consumption is divided into 144 time-slots of 10min each. The conventional user

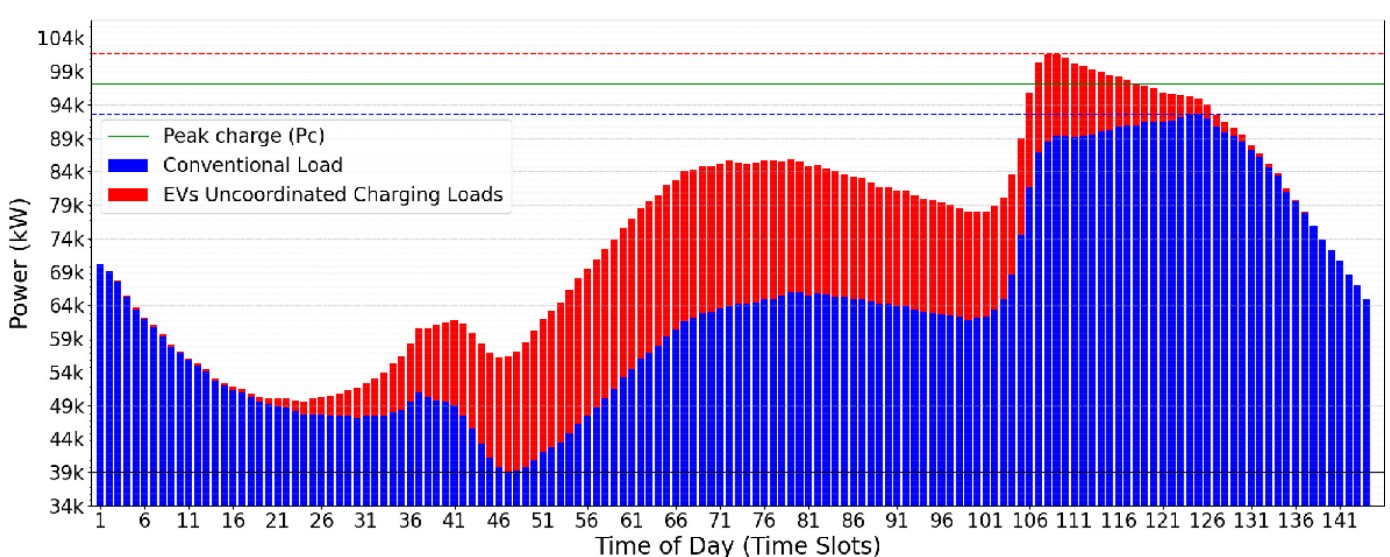

**Fig 2. Conventional load demand (blue) and EV load demand (red).**

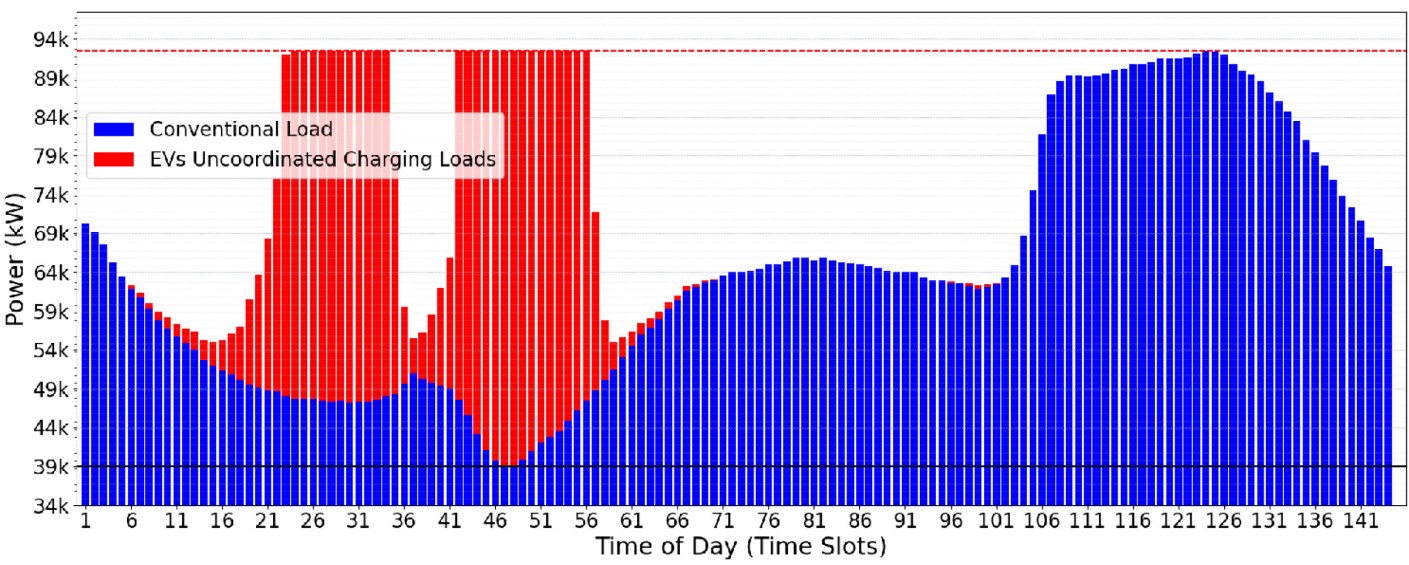

**Fig 3. Result found at the end of CVF.**

consumption (in blue) and the additional EVs power charging demands (in red) may be observed. The greedy load allocation strategy of CVF provides, as a result, the oscillatory allocation behavior shown in Fig 3. This behavior is considered harmful to the electrical power supply and the grid's stability.

Even in distributing the load of electric vehicles and effectively reducing the peaks of the electrical grid during peak hours, the CVF approach has some significant limitations. First, the algorithm requires rework in each iteration to find and fill up the valleys, which may result in operational inefficiencies. Furthermore, the initial $Pc$ value can be too high and should be optimized to improve system efficiency. However, the most serious problem arises when introducing an oscillatory profile into the electrical grid. This fluctuation can compromise the stability and reliability of the power supply, requiring a more sophisticated approach to managing the charging of electric vehicles.

### 3.2 Optimistic valley-filling (OVF)

Optimistic Valley-Filling (OVF) is a heuristic approach that changes the power peak charge ($Pc$) constraints for recharging electric vehicles. In contrast to the classic approach, where the $lowerPc$ is based on the maximum conventional electrical demand, OVF adopts the lowest value of this demand as the reference so that the new $Pc$ calculated by OVF is initialized with a more optimistic value. OVF does not change the classical algorithm itself (as presented in Algorithm 2) but instead adjusts the $Pc$ limits ($upperPc$ and $lowerPc$) as presented in Eq 2. As a result, the initial $Pc$ values found by OVF are lower than those determined by CVF (Fig 3), offering a more aggressive approach towards a minimum value of the $Pc$.

$$
\begin{aligned}
Pc &= \frac{upperPc + lowerPc}{2} \\
&= \frac{\boldsymbol{max}(EVsLoad + Conv.Load) + \boldsymbol{min}(Conv.Load)}{2}
\end{aligned}
\tag{2}
$$

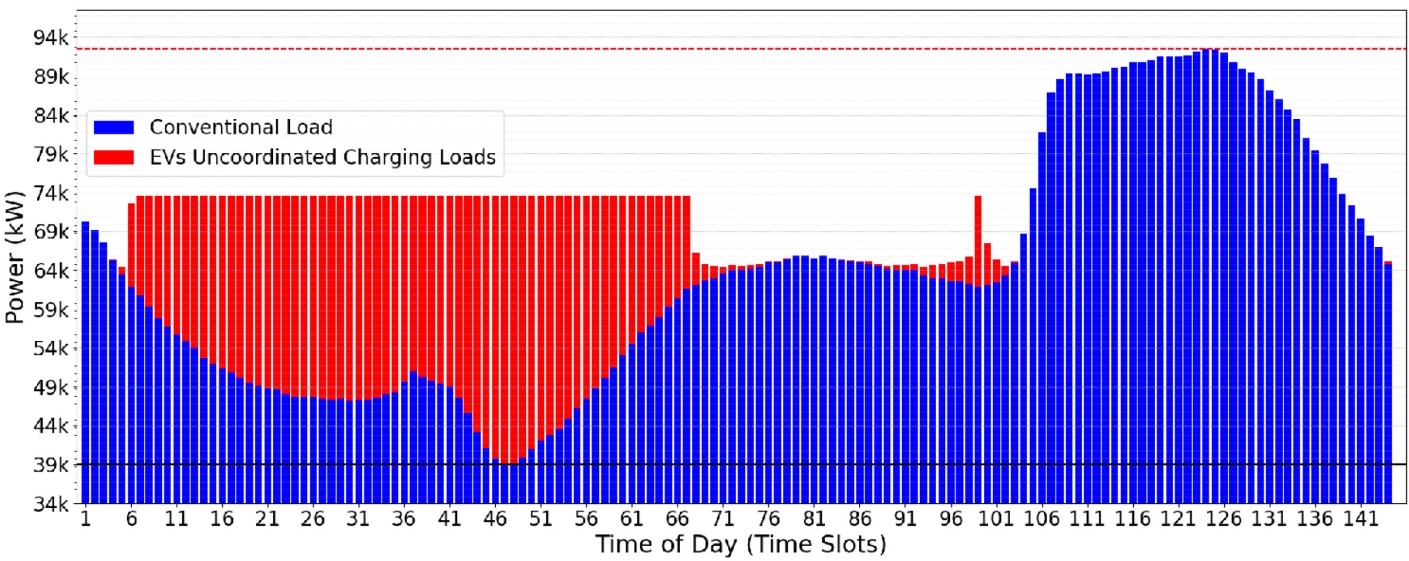

**Fig 4. Result of the OVF heuristic.**

The impact of the $Pc$ calculated by the OVF heuristic can be observed when comparing Figs 3 and 4. In the OVF algorithm, this initial value is considerably lower, which implies that fewer electric vehicles can be served in the first iteration. Starting with a lower $Pc$, this approach requires a slightly greater number of iterations to achieve convergence and meet the charging demand of electric vehicles. The benefit lies in eliminating the potentially oscillatory behavior that may happen when running CVF. Fig 4 presents the results of the OVF heuristic when allocating the EV charging demands informed in Fig 2. One may notice the lower operational effort and greater stability in the electrical grid when compared Figs 3 and 4.

The oscillatory behavior in the Classical Valley-Filling (CVF) approach, marked by sharp allocation spikes across time slots (TS), causes significant variations in grid stability. Specifically, during high-demand periods, CVF allocations result in demand fluctuations of up to 15% between consecutive TS, placing additional strain on the grid's balancing mechanisms. This oscillatory load pattern, illustrated by demand fluctuations in peak power requirements in Fig 3, highlights the challenge of maintaining a consistent load distribution. In contrast, the OVF heuristic mitigates these fluctuations by allocating charges more gradually and with less deviation, resulting in a more stabilized demand curve, as shown in Fig 4. This quantitative comparison highlights the OVF approach's suitability for applications requiring enhanced grid stability and smoother load transitions.

By initiating the $Pc$ with a more optimistic value, OVF promotes an approach towards a lower $Pc$ value, thus mitigating the stress of the electrical grid. The OVF represents a promising strategy for optimizing electric vehicle charging systems while ensuring the reliability and efficiency of the grid.

## 4 Load conservation valley-filling (LCVF)

This section outlines the methodology employed to the design of the Load Conservation Valley-Filling (LCVF) heuristic. The primary objective of LCVF is to optimize electric vehicle (EV) charging schedules within periods of lower conventional energy demand, reducing peaks and ensuring stability in the electrical grid. The methodology addresses the limitations

observed in previous valley-filling (VF) approaches, specifically Classic Valley-Filling (CVF) and Optimistic Valley-Filling (OVF).

In the VF approach, a 24-hour daily interval is divided into fixed time slices, where EV charging demands are strategically allocated to TS with lower conventional consumption (valleys). However, CVF and OVF heuristics present challenges: CVF may introduce oscillatory behaviors in the power allocation process, leading to grid instability, while OVF, although less prone to oscillations, does not fully address the reallocation redundancies that compromise computational efficiency.

The design of this new approach incorporates a memory-preserving mechanism, allowing the algorithm to retain information about successfully allocated EV loads across iterations. This feature ensures that the allocation state of previously scheduled EVs is maintained, reducing redundant operations in future iterations. The details of the LCVF heuristic are described in the following subsection, providing a comprehensive understanding of the modifications applied to the classic VF framework.

## 4.1 Methodology

To facilitate understanding of the Load Conservation Valley-Filling (LCVF) heuristic, Fig 5 presents a flowchart outlining the key steps in its execution. The flowchart provides a visual representation of how LCVF optimizes the electric vehicle (EV) charging process by iteratively identifying valleys in the power consumption profile, preserving allocation states, and adjusting the power constraints according to the grid's load capacity.

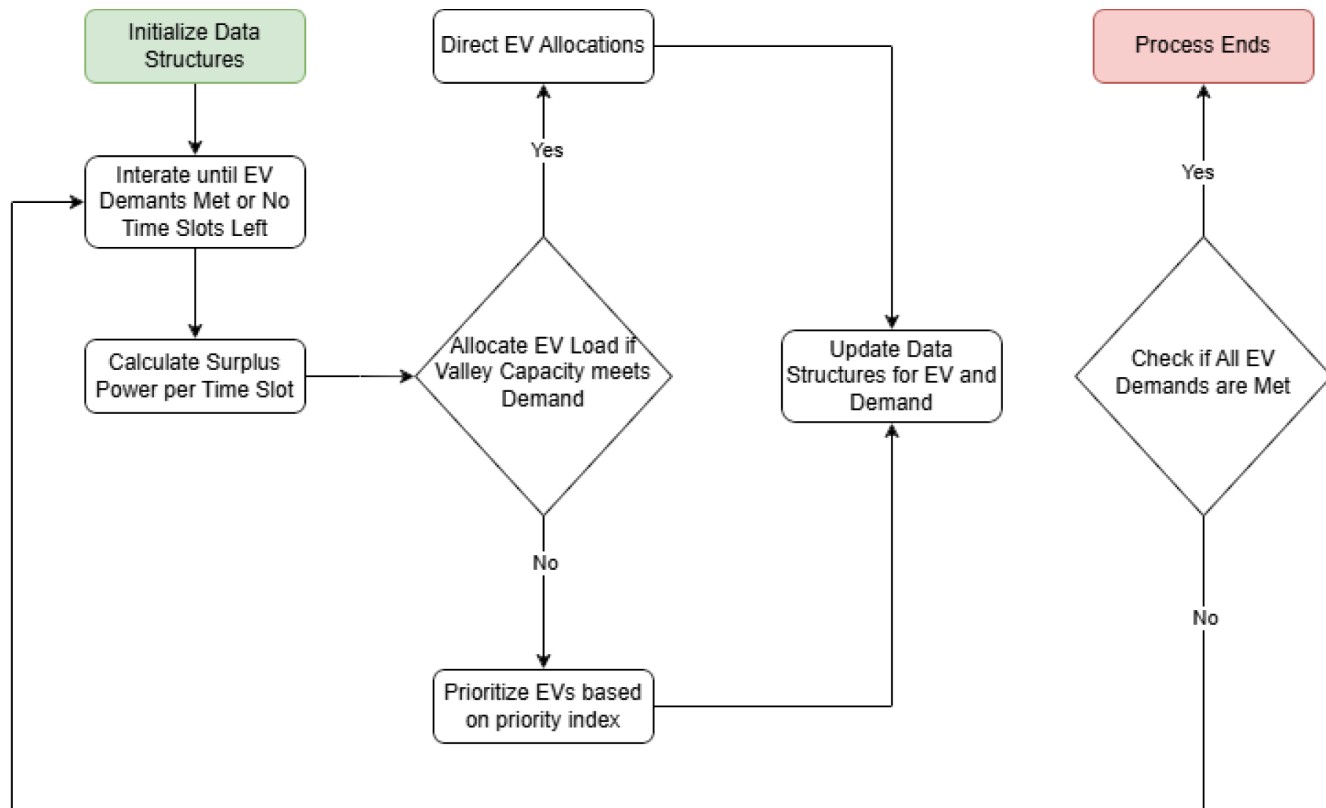

**Fig 5. Load conservation valley-filling (LCVF) heuristic flowchart.**

The flowchart begins with the initialization of data structures for both conventional demand and EV load data. Then, it enters the main iteration loop, which repeats until all EV demands are met or no additional TS are available for allocation. In each iteration, the algorithm performs the following steps:

1. **Calculate Surplus Power:** For each TS, calculate the surplus power by subtracting conventional demand from the current peak charge limit ($Pc$).

2. **Identify Valley:** Using the surplus power values, calculate a capacity margin index and identify the valley with the highest margin, where EVs will be allocated.

3. **Allocate EV Loads:**

   - If the valley capacity meets the demand, perform a direct allocation.

   - If the capacity is insufficient, prioritize EVs based on a priority index and allocate demands accordingly.

4. **Update Data Structures:** After allocation, update the conventional demand profile by storing the allocated EV loads, and exclude fully charged EVs from subsequent iterations.

5. **Check Completion:** Verify if all EV demands are met. If so, the process ends; otherwise, repeat with adjusted $Pc$ values.

This structured approach minimizes reallocation redundancies and stabilizes the electrical grid load profile. The memory-preserving aspect ensures that previously allocated slots are not revisited, optimizing computational efficiency and maintaining grid stability.

## 4.2 Design principles behind LCVF

The Load Conservation Valley-Filling (LCVF) heuristic keeps information about EV load allocation in each iteration. Its primary purpose is to promote the stability of the electrical grid, encompassing both conventional consumption and EV charging. This heuristic reshapes the greedy approach of the classical algorithm, mitigating fluctuations introduced into the power grid.

The "state storage" concept in the LCVF algorithm is a key component in its efficiency and stability improvements compared to CVF. This storage mechanism allows LCVF to retain information on allocated EV loads from one iteration to the next, avoiding redundant calculations and reallocations. By preserving the charging state of vehicles that have already been successfully allocated, the algorithm bypasses re-processing these vehicles in subsequent iterations, thereby reducing computational time and load oscillations on the grid.

Unlike CVF, which recalculates and reallocates EV loads without considering prior iterations, LCVF's state storage ensures a cumulative allocation pattern. This approach minimizes fluctuations in power demand by keeping already allocated loads stable across iterations, which ultimately enhances grid stability. State storage allows LCVF to focus computational resources only on unallocated EV demands, contributing to faster convergence to an optimized load distribution.

To visually represent this, Fig 6 demonstrates the iterative process in LCVF, illustrating how EV load data is preserved across iterations. This visual comparison highlights how state storage progressively reduces the computational effort required and stabilizes demand, underscoring the algorithm's advantage in maintaining a balanced grid load over time.

The main LCVF algorithm uses the same procedure to calculate the $Pc$ value of the OVF heuristic. The main difference between LCVF and Algorithm 2 is the fact that in line 4, there is

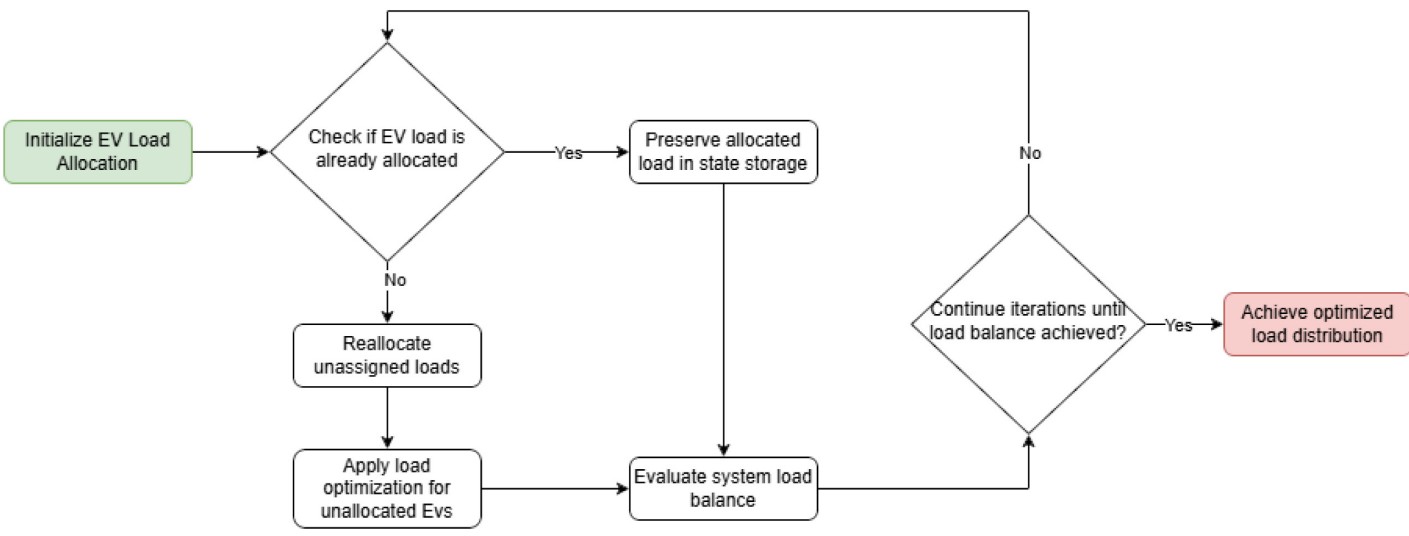

**Fig 6. Schematic flow of the state storage process in the LCVF algorithm.**

a return of three parameters and not just one, as in the previous cases. When calling the function that performs LCVF, the algorithm will return the Flag indicating if the *Pc* can be decreased or increased (*Flag* = 1 or *Flag* = 0), indicating whether all the demands of the electric vehicles were met or not, and the updated datasets *dataEn* and *dataEv*.

The steps carried out by LCVF are presented in Algorithm 3 where the highlighted lines (blue) represent the new steps introduced in this approach. The algorithm starts by initializing the necessary data structures. It first creates copies of the current energy data (*dataEn*) and the current EV data (*dataEv*) by setting *dataEn* to *En* and *dataEv* to *Ev* (lines 2-3). This copy guarantees that the original data can be restored (lines 52-55) if the electric vehicle recharging demand is not met and the *Pc* value should be increased.

The algorithm main loop (line 4) runs while there are still available TS (*sizeof*(*En.TS* > 0)). Within each iteration, it calculates the surplus power for each TS *k* (line 5-7) by subtracting the conventional power consumption at TS *k* (*En.Pcon$^k$*) from the peak charge value (*Pc*). This surplus power (*En.Psup$^k$*) is a key metric for determining the additional power allocated to an EV charging without exceeding the power peak limit.

Lines 8-14 show that the algorithm calculates the total demanded power for each TS. For each EV *n* that is within its charging window, its start time (*Ev.Start$_n$*) is less than or equal to the current TS *k* and its end time (*Ev.End$_n$*) is greater than or equal to *k*, the demanded power (*En.Pdem$^k$*) in TS *k* is incremented by the power required by EV *n* (*Ev.P$_n$*).

Lines 15-17 set the capacity margin index (*En.IndexTS$^k$*) for each TS *k* by dividing the surplus power (*En.Psup$^k$*) by the total demanded power (*En.Pdem$^k$*). This index allows to identify the TS where the grid has the most capacity to handle additional charging loads. Line 18 selects the TS *H* with the highest capacity margin index, thus identifying the time-slot which is the valley in the power consumption profile.

Lines 19-27 allocate the EV charging loads based on the calculated indices. If the margin index for the selected TS *H* is greater than or equal to 1, the algorithm directly allocates the charging demand of vehicle *n*. For each electric vehicle *n* within its charging window, it is checked whether the vehicle still demands energy (*Ev.E$_n$* > 0) and whether the same vehicle has not yet occupied TS *H* in previous allocations (*H* $\notin$ *Ev.Block$_n$*). This blocking is necessary

due to the maximum recharging limitations of each TS, considering that once the vehicle is allocated to a TS, it will use the maximum resources of that charger. If the charging conditions are satisfied, the accumulated charging energy of $H$ ($En.Pacc_n^H$) is increased by the energy required by the electric vehicle $n$ ($Ev.P_n$). Additionally, the vehicle's required energy ($Ev.E_n$) is decreased by the allocated energy, the remaining charging time ($Ev.T_n$) is decreased, and $H$ is added to the list of time intervals blocked to vehicle $n$ ($Ev.Block_n$).

If the margin index is less than 1 (line 28), the algorithm calculates the charging priority index for each electric vehicle within its charging window (lines 29-35). The new LCVF approach did not change this code block. The charging priority index ($Ev.IndexEv_n^H$) is determined by dividing the vehicle's remaining energy requirement ($Ev.E_n^H$) by the product between the remaining charging time ($Ev.T_n^H$) and the power required by the vehicle ($Ev.P_n$). The index is set to zero if the vehicle does not meet its loading window (line 33). In line 36, electric vehicles are ordered (using a quicksort ordering algorithm) according to their priority indexes.

The algorithm then allocates (lines 37-46) the charging loads based on this priority. For each vehicle $n$ in the ordered list, if the excess energy for the TS $H$ ($Ev.Psup^H$) minus the energy required by the vehicle ($En.P_n$) is greater than or equal to zero and the vehicle is not using the TS ($H \notin Ev.Block_n$), the accumulated charging energy of the vehicle is increased by $Ev.P_n$, and the excess energy is decreased by $Ev.P_n$. Furthermore, the required energy ($Ev.E_n$) is decremented by the allocated energy, the remaining charging time ($Ev.T_n$) is decremented, and the TS $H$ is added to the TS blocked list of the vehicle ($Ev.Block_n$).

At the end of each iteration (line 47), TS $H$ is removed from the set of available TS for electric vehicle charge allocation ($E_n.TS-\{H\}$). After iterating for all TS, the algorithm checks if all EVs have met their energy demands (lines 49-50). If all EVs can be fully charged, $Flag = 1$ indicates completion. Otherwise (lines 51-55), $Flag = 0$ and the total conventional energy consumption ($En.Pcon$) are updated by adding the accumulated charging energy ($En.Pacc$). As the $Pc$ will need to be increased to accommodate the vehicle demands, the data structures are updated ($dataEn = En$ and $dataEv = Ev$) to preserve the modifications made in this iteration for future iterations.

**Algorithm 3** Load Conservation Valley-Filling (LCVF)

```
 1: function LCVF_SCHEDULING(dataEn, dataEv, Pc)
 2:    En = dataEn
 3:    Ev = dataEv
 4:    while sizeof(En.TS) > 0 do
     # Calculate surplus power for each TS
 5:       for each TS k ∈ En.TS do
 6:         En.Psup^k = Pc − En.Pcon^k
 7:       end for
     # Calculate total demanded power for each TS
 8:       for each TS k ∈ En.TS do
 9:         for each EV n ∈ Ev.N do
10:           if (Ev.Start_n ≤ k) and (k ≤ Ev.End_n) then
11:             En.Pdem^k = En.Pdem^k + Ev.P_n
12:           end if
13:         end for
14:       end for
     # Calculate capacity margin index for each TS
15:       for each TS k ∈ En.TS do
16:         En.IndexTS^k = En.Psup^k/En.Pdem^k
17:       end for
     # Select the index with the highest margin index (valley)
18:       H = index(max(En.IndexTS))
19:       if En.IndexTS^H ≥ 1 then                              ▷ Case 1
```

```
        # Allocate charging loads directly and update EVs
20:        for each EV n ∈ Ev.N do
21:          if (Ev.Start_n ≤ H) and (H ≤ Ev.End_n) and (Ev.E_n > 0) and
    (H ∉ Ev.Block_n) then
22:            En.Pacc_n^H = En.Pacc_n^H + Ev.P_n
23:            Ev.E_n = Ev.E_n − Ev.P_n^H
24:            Ev.T_n = Ev.T_n − 1
25:            Ev.Block_n = Ev.Block_n + {H}
26:          end if
27:        end for
28:      else                                              ▷Case 2
      # Calculate charging priority index for each EV
29:        for each EV n ∈ Ev.N do
30:          if (Ev.Start_n ≤ H) and (H ≤ Ev.End_n) then
31:            Ev.IndexEv_n^H = Ev.E_n^H / (Ev.T_n^H × Ev.P_n)
32:          else
33:            Ev.IndexEv_n^H = 0
34:          end if
35:        end for
      # Sort vehicles according to priority index
36:        IndexSort ← qsort(Ev.IndexEv)
      # Allocate charging loads based on priority and update EVs
37:        for each EV n ∈ IndexSort do
38:          if (Ev.Psup^H − En.P_n ≥ 0) and (H ∉ Ev.Block_n) then
39:            Ev.Pacc_n^H = Ev.Pacc_n^H + Ev.P_n
40:            En.Psup^H = En.P_sup^H − Ev.P_n
41:            Ev.E_n = Ev.E_n − Ev.P_n^H
42:            Ev.T_n = Ev.T_n − 1
43:            Ev.Block_n = Ev.Block_n + {H}
44:          end if
45:        end for
46:      end if
47:      En.TS = En.TS − {H}
48:    end while
49:    if sum(Ev.E) == 0 then
50:      Flag = 1
51:    else
      # Update data for each TS used in the current Pc allocation
52:      Flag = 0
53:      En.Pcon = En.Pcon + En.Pacc
54:      dataEn = En
55:      dataEv = Ev
56:    end if
57:    return dataEn, dataEv, Flag
58: end function
```

One may be noticed that the algorithm works from new data structures. The grid profile data and the EV demands data are updated to take advantage of the allocated power already achieved in the following steps. If the *Pc* satisfies the demand for electric vehicles, its value is reduced, and a new allocation of electric vehicles is run without preserving the previous allocation.

The CVF algorithm focuses exclusively on allocating the EV charging demands to periods of lower power usage in each iteration. CVF does not incorporate any state storage from one iteration to another. This lack of memory between iterations results in performing repeated procedures all over again when the valley-filling core is run, retracing the same steps, and finding the same values. This is because the energy distribution of conventional demand is not

modified along the schedule. The OVF heuristic, despite using an optimistic $Pc$ approach and solving network oscillation problems, also faces the problem of rework.

The main motivation for designing the LCVF heuristic is based on the idea that storing the allocation state of EVs that are already fully allocated allows the algorithm to identify new valley points. The goal is to make the total consumption curve of the electrical grid as stable and flat as possible, avoiding rework caused by the allocation of EV demand at each iteration. Starting from an initial state, as described in Fig 2, it is possible to explore better the valleys filling up to the delimiter $Pc$, which is gradually increased at each iteration, resulting in a flatter EV load allocation, as represented in Fig 7. The information (current status of the EV load at each time interval) is preserved between iterations, preventing the same valley points from being found repeatedly. By storing information about the vehicles allocated between iterations, the number of vehicles remaining for scheduling is expected to reduce gradually. The additional time required to store data in each interation is felt only in the first iterations since the algorithm will have fewer elements to allocate in later iterations.

The memory element affects the conventional demand curve in subsequent iterations, recording the loads allocated to the valley in the current iteration. Another crucial aspect is that vehicles that have already been scheduled must be excluded from the dataset to be analyzed in the next iteration, which also requires the storage of a list containing the time intervals in which each vehicle remained connected to the electrical grid. If an electric vehicle is allocated to a valley and its charge has already modified the conventional curve for the next iteration, that time interval is removed from the set, and the same vehicle cannot be assigned to that interval in an iteration ahead.

The computational complexity of the LCVF algorithm is primarily driven by the main loop, which iterates over all time slots ($T$). Within each iteration, operations may traverse all electric vehicles ($N$). The most computationally expensive steps include calculating the demanded and surplus power for each time slot and vehicle, resulting in an overall complexity of $O(T^2 \times N)$. Although the algorithm contains a sorting step with $O(N \log N)$ complexity, the execution time is mainly dictated by the number of time slots and vehicles, preventing an overall logarithmic complexity.

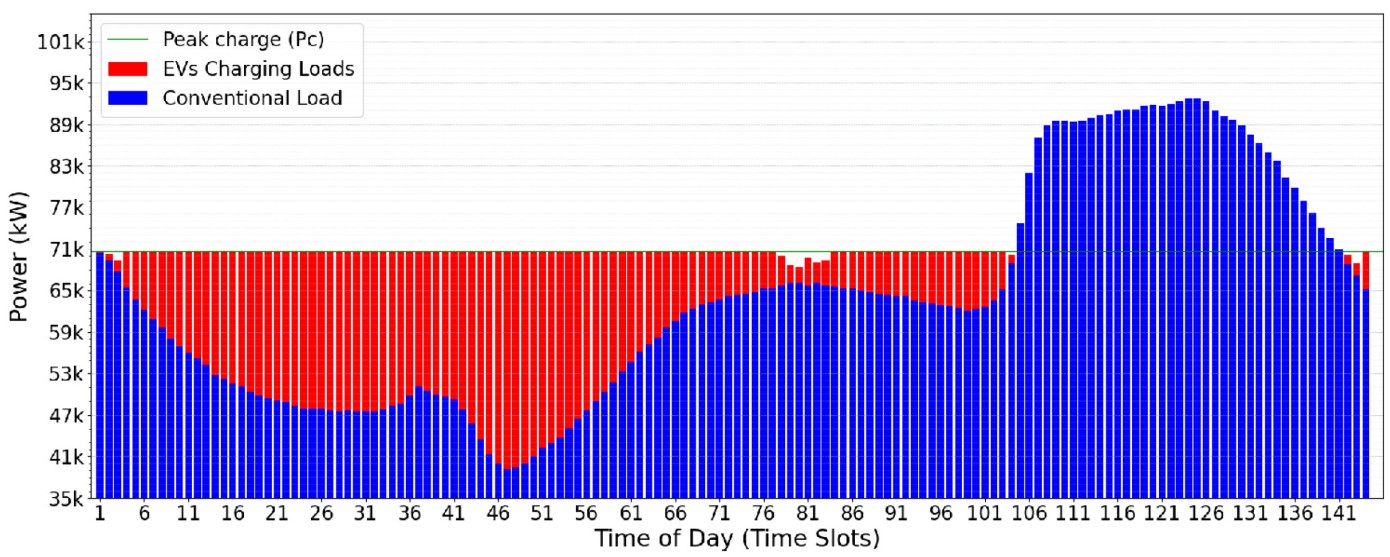

**Fig 7. Result of the LCVF heuristic.**

The LCVF algorithm is designed to allocate charging loads by prioritizing the minimization of demand peaks. This approach systematically distributes the EV charging sessions into valley periods, reducing demand spikes while conserving energy. Unlike the Classic Valley-Filling (CVF), which evenly fills demand valleys without specific load conservation considerations, the LCVF actively adapts to fluctuating grid conditions to maintain lower peak demands. The Optimistic Valley-Filling (OVF) algorithm, meanwhile, adopts a more flexible approach, allocating loads primarily during off-peak times but without the same level of peak control, making it suitable for scenarios with low variability in demand.

## 5 Use-case scenarios

Experiments were conducted to evaluate the three heuristics across multiple EV charging and load management scenarios, using the Tetouan power consumption dataset, originally compiled by Salam and El Hibaoui [25]. The dataset includes hourly records of power demand alongside environmental variables such as temperature, humidity, and wind speed, capturing typical urban energy consumption patterns. Subdivided into 10-minute intervals, this dataset is well-suited for input to the VF heuristics; a single preprocessing step was applied to remove variables not used by valley-filling techniques. The fields in this dataset include the following:

- *Date*: the day of the measurement.

- *Time*: time the measurement was taken.

- *TS*: Time-slots were assigned for each measuring time.

- *PConvTotal*: Total conventional demand for electrical energy requested by the city consumers.

The 10-minute intervals in the Tetouan dataset are optimal for the VF approach, providing fine granularity that enables precise adjustments in EV charging load allocation. This interval duration is short enough to capture fluctuations in conventional energy demand throughout the day, enhancing the ability to identify valleys accurately and distribute EV charging demands without exceeding the peak charge limit.

Electric vehicles' charging/recharging data is from the "City of Boulder Open Data" [26] dataset. This dataset contains $50k$ EV charging/recharging samples from 2018-2023. Like the previous dataset, the EV data had a preprocessing step to remove some fields. The variables used in the experiment are the following:

- *N*: EV Identification.

- *Start*: The TS when the vehicle is connected to the power grid.

- *End*: Time slot in which the vehicle is disconnected from the electrical network.

- *Power*: Power demand per TS of the EV.

- *Energy*: Energy required to recharge the EV fully.

- *TSRemaining*: Number of available TS for the EV to remain connected to the grid.

Some additional modifications were also necessary for the fields used for the experiments. Among these adaptations, grouping the charging/recharging data was necessary. As a few units of electric vehicles are being recharged daily, this consumption needs to be more significant to impact the city's conventional consumption. Therefore, we group the charging/recharging measurements to correspond to a single day of recharge. When considering the

entire demand for electric vehicles in the data set representing a 24-hour interval, it is possible to observe the increase in energy in the traditional curve, bringing impacts to the power grid.

## 5.1 Original scenario

The Original Scenario represents a base case in which time slots for EV charging are tightly constrained, limiting flexibility. This scenario establishes a benchmark for assessing the potential of the CVF, OVF, and LCVF approaches under strict time restrictions. The original data from the two datasets create a scenario in which few TS are available to complete the recharging of electric vehicles. This feature made recharging periods less flexible, with minimal scheduling possibilities. Fig 8 illustrates conventional and electric vehicles' daily energy consumption data from the Tetuan and Boulder datasets.

In this first scenario, the time electric vehicles remained connected to the grid was very close to the time needed to recharge. Therefore, the heuristics could not significantly change the scheduling, as there were not enough time margins for these changes in the vehicle connection. This scenario imposes strong constraints on the VF heuristics to schedule the demands of electric vehicles, thus impacting the $Pc$ values and the runtime.

## 5.2 Flexible scenario

The Flexible Scenario introduces additional connection time for EVs, providing increased scheduling options. This scenario allows us to observe the adaptability of each heuristic in contexts with moderate flexibility. In the second user-case scenario, an increase in the connection time of electric vehicles is observed, allowing greater flexibility in scheduling the charging time. The electric vehicle dataset has been modified to provide a larger margin in the charging period intervals when the vehicle is connected to the grid. More time of connection means that electric vehicles now have more time available on the electricity grid, being able to choose the time they need to be recharged or remain connected to the grid without actually consuming energy. The original values of the start and end variables from the electric vehicle dataset were randomly changed (using a fixed seed to ensure replication of results), providing 20 additional TS before and after connecting the electric vehicle to the grid.

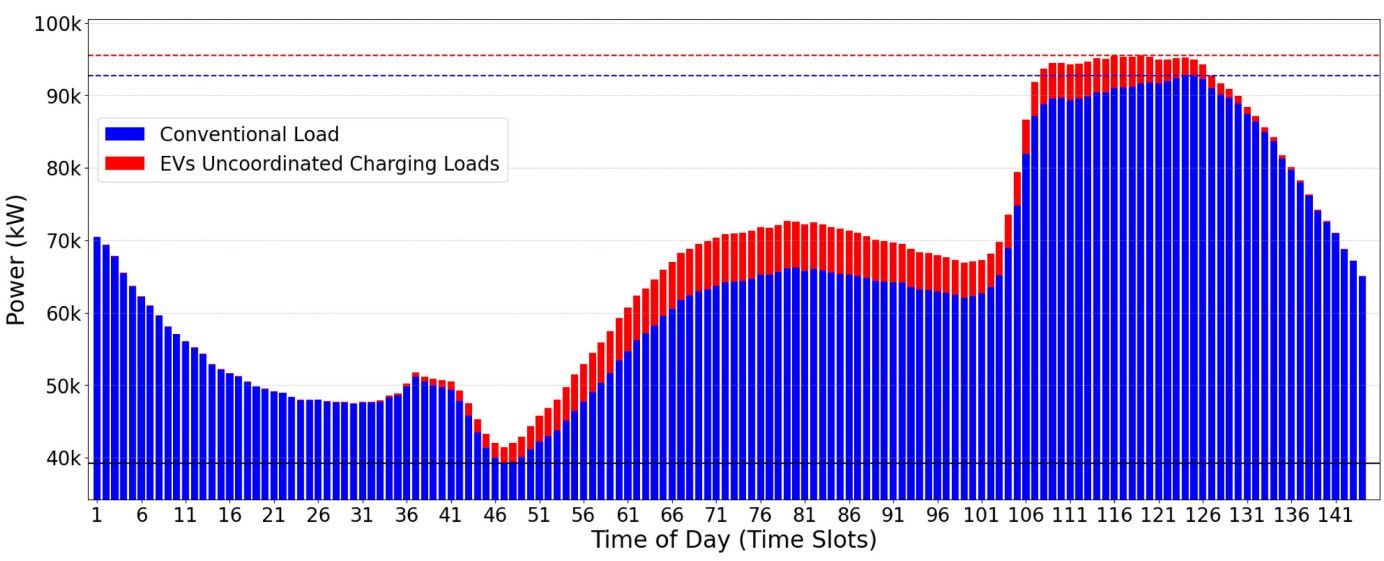

**Fig 8. Original (first use-case) scenario: Daily power demands of conventional and electrical vehicle.**

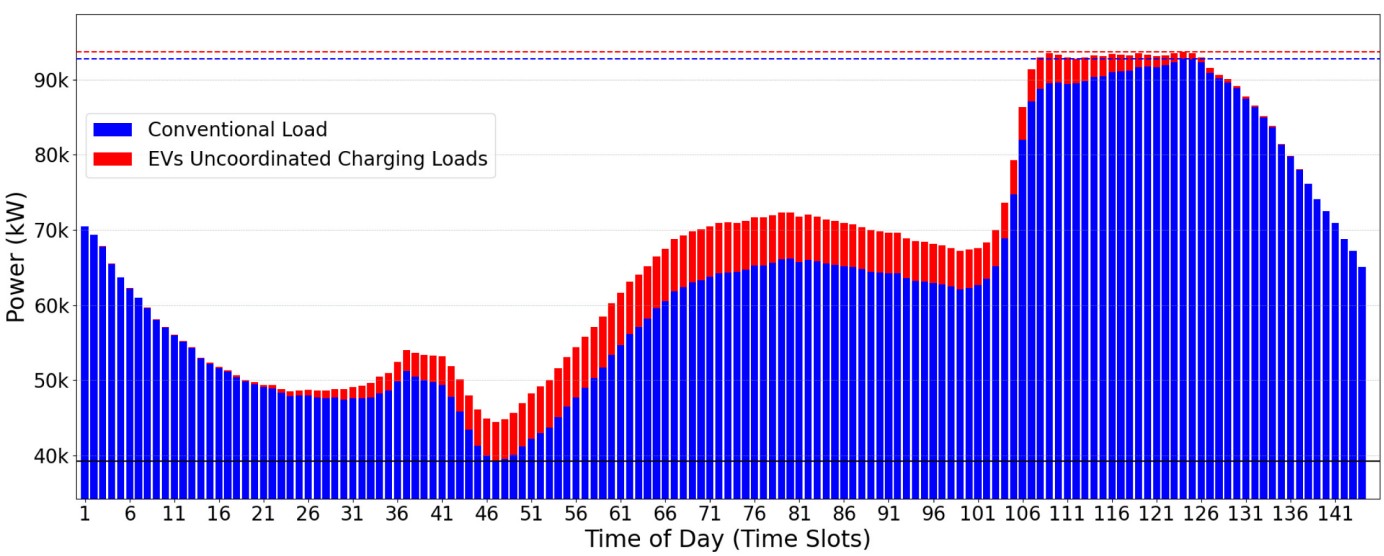

**Fig 9. Flexible (second use-case) scenario: Daily power demands of conventional and electrical vehicle.**

Fig 9 shows the flexible scenario with conventional consumption and EV data. The figure shows a change in peak hours of electricity consumption. The introduced flexibility reduces total electricity demand during peaks compared to the previous scenario. Introducing additional periods before and after connecting vehicles to the grid allows for a more balanced distribution of energy demand over time.

Extending the connection time of EVs is an alternative strategy for reducing consumption peaks and relieving pressure on the electricity grid. This flexibility improves resource management and reduces operational costs. The approach promotes greater stability and minimizes grid tension, adjusting recharge time according to energy availability and individual needs.

## 5.3 Increased consumption

The Increased Consumption Scenario simulates a high-demand environment by raising the energy consumption factor for each EV. Here, the algorithms' capacity to allocate demands effectively under pressure is evaluated. Based on the flexible scenario (second use case), the electrical energy demand needed to recharge the vehicles still does not significantly impact the grid. In other words, the consumption peaks generated by electric vehicles need to be higher to stress the energy distribution grid. A viable alternative for simulating a worst-case situation would be to increase the EVs' consumption and evaluate the impact of this change.

To increase the total peak energy demand and aggravate the challenge of integrating electric vehicles into the grid, we increase the consumption of each vehicle by a factor of 3.5×. This value was chosen to examine the performance of the heuristic in a scenario with high power demand but offering high timing flexibility. A higher factor was not used since it could significantly distort the energy curve required, making it difficult to visually compare the scenarios and generating a disparity in the results (Fig 10).

As the consumption of electric vehicles increases, the total peak in electrical energy consumption attributed to them becomes more evident. Furthermore, as all vehicles have proportionally increased their consumption, the points with the highest concentration of electric vehicles are more evident, as illustrated in Fig 10. This uniform increase in consumption

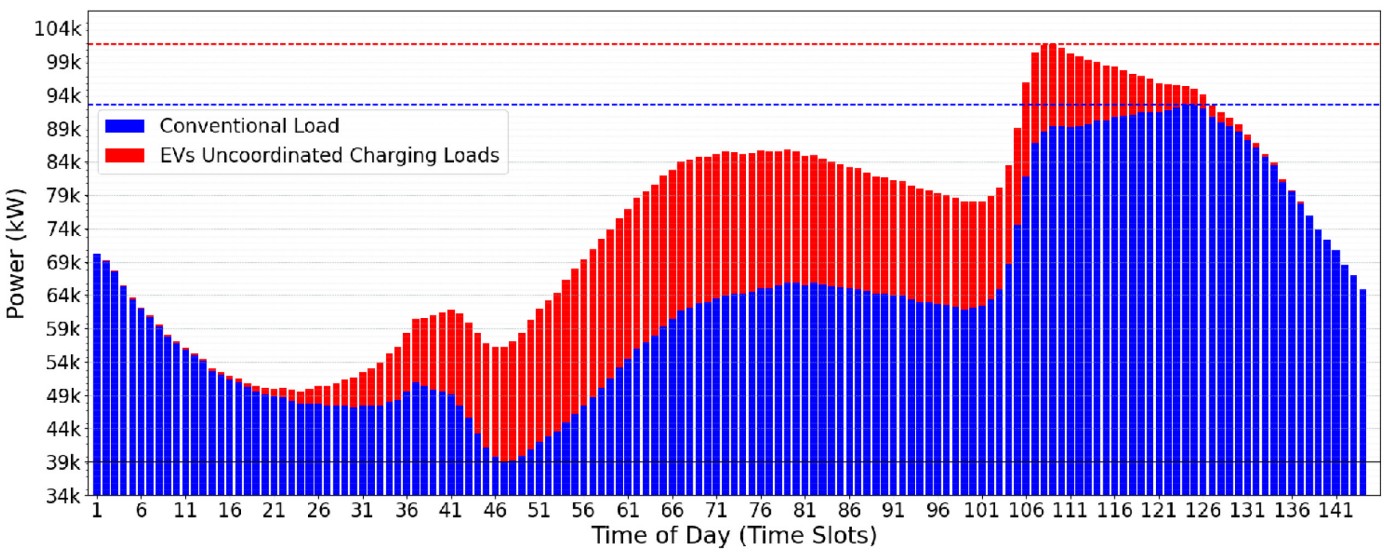

**Fig 10. Increased EV power demand (third use-case) scenario: Daily power demands of conventional and electrical vehicle.**

further highlights the power demand for vehicles, demonstrating more clearly the impact they can have on the electricity grid.

### 5.4 8-hour availability

The 8-Hour Availability Scenario expands the potential charging intervals by providing a dedicated 8-hour period for EVs to remain connected to the grid. This setup allows for assessing each heuristic's efficiency in utilizing grid resources under moderately flexible charging conditions. A fourth use-case scenario was developed to model the time intervals in which vehicles are typically parked, covering various environments such as work hours, school shifts, and other business periods. Within this scenario, each vehicle must remain connected to the electrical grid for at least 8 hours. By guaranteeing this prolonged connection time, the scenario establishes an interval with multiple available TS, allowing efficient scheduling of charging sessions for each vehicle.

Each vehicle can be recharged during at least one of the three 8-hour periods of the day. To distribute recharges over these three shifts, all vehicle data were modified to connect between TS 1 and 48 and had the '*Start*' field adjusted to TS 1, allowing them to be recharged at any time within the range of 1 to 48. Likewise, for the '*End*' field, any value between 1 and 48 receives the final TS 48, corresponding to the last unit of time available in that interval.

Fig 11 shows the 8-hour availability of EV power demand. The background colors represent these shifts of 8 hours each. Two ranges can also encompass one EV if one of its TS ('*Start*' or '*End*') crosses the border and is included in another value range.

The scenario accommodates overlapping TS, promoting more flexible EV charging and enhancing the use of grid resources. This scenario generally improves scheduling and resource utilization, increasing efficiency and reliability in electric vehicle charging systems.

### 5.5 24-hour availability

The 24-Hour Availability Scenario extends the charging interval availability to a full 24-hour period, offering maximum flexibility for scheduling EV loads. This scenario is designed to

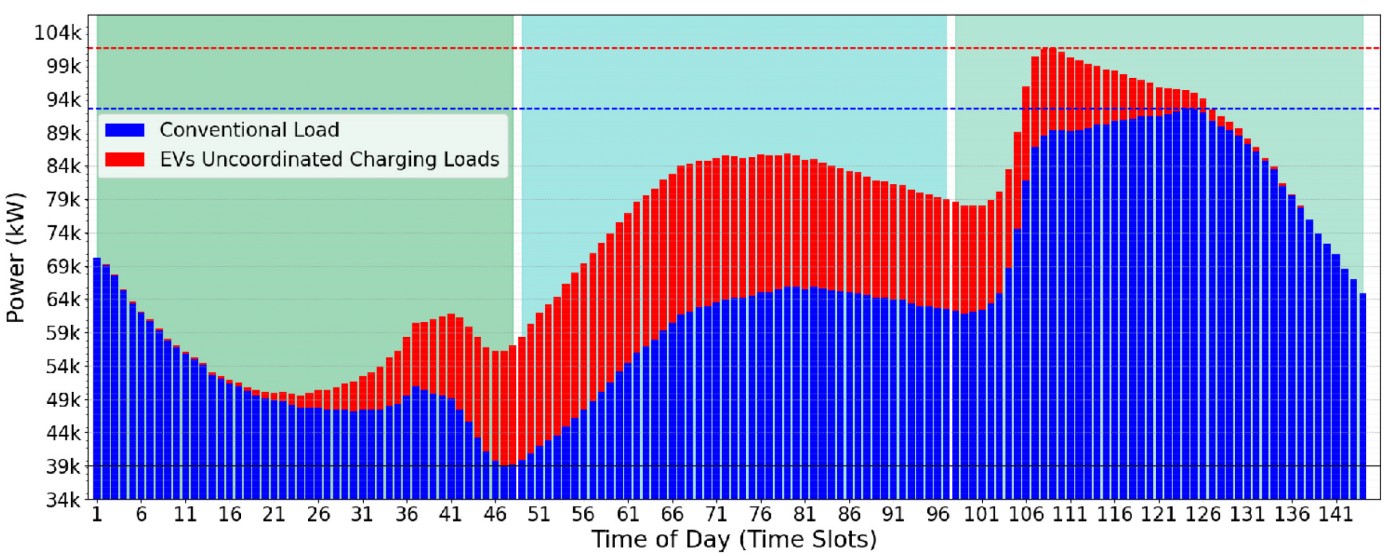

**Fig 11. 8-Hour availability EV power demand (fourth use-case) scenario: Daily power demands of conventional and electrical vehicle.**

evaluate each heuristic's capacity to optimize resource use and stabilize grid demand when given extensive charging freedom. The last use-case scenario was designed to simulate the context in which the electric vehicle is available all day for recharging. In this scenario, electric vehicles have 24 hours to schedule their power demand. This scenario is often used in airports and bus station yards, where vehicles remain parked for long periods, often lasting more than one day. Considering the conventional daily demand from the electricity grid, we have set an upper bound period of 24 hours.

In order to adapt the data to this scenario, all TS in which the vehicles are connected were modified. For all vehicles, the '*Start*' field is adjusted to the first TS (i.e., TS 1), and the '*End*' field is set to the last TS (i.e., TS 144). In Fig 12, the total filling of the bottom of the curves represents the complete availability of electric vehicles during the daily period.

Extending the connection duration provides the heuristics with increased scheduling flexibility, enabling them to achieve a more stable total energy demand curve. This modification allows the algorithm to manage energy supply and demand effectively by filling demand gaps in off-peak hours and efficiently allocating energy resources.

## 6 Results and discussions

In this section, we present experiments and results to validate and evaluate the LCVF heuristic by comparing it to two other approaches, CVF and OVF, in each of the five use-case scenarios. In the experiments, we determined that each heuristic should end its execution when the value of the unallocated load reached zero or when the absolute percentage difference of the *Pc* parameter (the limiting variable for load allocation of electric vehicles) was less than 1% between iterations of the algorithms. The experiments were performed on the Google Colaboratory platform, running a Jupyter Notebook framework service. The heuristics were implemented using the Python language version 3.10.12 and Linux Ubuntu 18.04.3 LTS. The runtime experiments were carried out on a dual-core machine: Intel® Xeon® CPU 2.20GHz, RAM 12.7 GB, and disk 107.7 GB. The datasets and heuristics used in the experiments and evaluations in this work are available at Souza et al. [27].

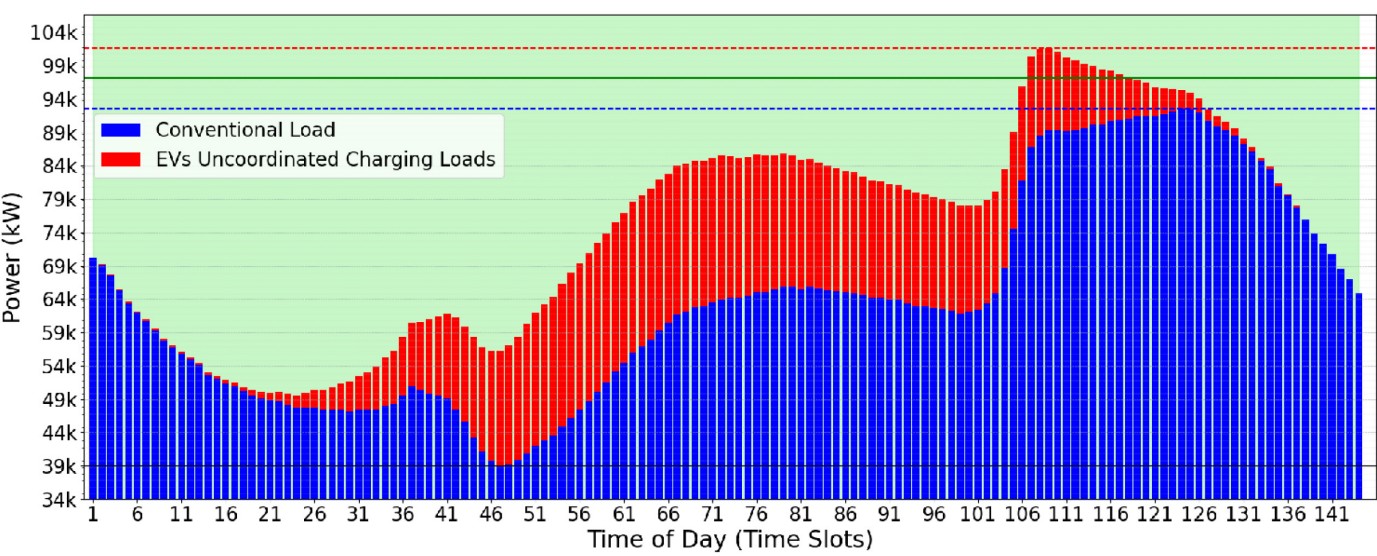

**Fig 12. 24-Hour availability EV power demand (fifth use-case) scenario: Daily power demands of conventional and electrical vehicle.**

## 6.1 Results on peak charge and runtime

In this study, a 1% convergence criterion was applied to balance computational efficiency and accuracy in the load allocation process, ensuring demand smoothing without excessive runtime. Testing showed that a stricter threshold, like 0.5%, increased runtime with minimal benefit, while a more lenient 10% threshold accelerated convergence but compromised load stability. The 1% criterion consistently produced stable results across the LCVF, CVF, and OVF heuristics, enabling efficient resource use and enhancing the LCVF's memory-preserving features by reducing reallocation redundancies. This convergence threshold thus supports the robustness and applicability of the LCVF approach in dynamic EV charging scenarios.

When evaluating the heuristics' performance in EV load charging scenarios, it is mandatory to observe two metrics: Peak power Charge (*Pc*) and Unallocated Load. After all heuristic iterations, the final value of *Pc* represents the amount (in Watts) of resources needed to meet electric vehicles' energy demands. The Unallocated load means the amount of EV charging power the heuristic cannot allocate in the TS. Reducing the curve for the Unallocated Load parameter ensures that its curve equals zero, which means all the electric vehicle demands are met.

Fig 13 displays the results of the experiments. The graphs on the left demonstrate the peak power required by each technique in each scenario, represented by the *Pc* parameter and the runtime. Similarly, the graphs on the right illustrate the Unallocated Load of EVs. The filled circles in the lines represent the interactions of each heuristic. One may observe that the Unallocated load shrinks according to the number of interactions, thus showing that all the techniques could meet the demands of the scenarios. The legend in each group of graphs assigns numbers and acronyms representing the evaluated techniques: Classic Valley-Filling (CVF), Optimistic Valley-Filling (OVF), and Load Conservation Valley-Filling (LCVF). The first number indicates the technique, and the second represents the scenario: 1—Original Scenario, 2—Flexible Scenario, 3—Increased Consumption, 4—8-Hour Availability, and 5—24-Hour Availability). The best result is achieved by combining the lowest runtime according to the stopping condition (Unallocated Load equal to 0 and percentage difference, less than 1%, between the *Pc* values of two sequential interactions) and the lowest resource allocation value for electric vehicles to ensure that everyone is recharged.

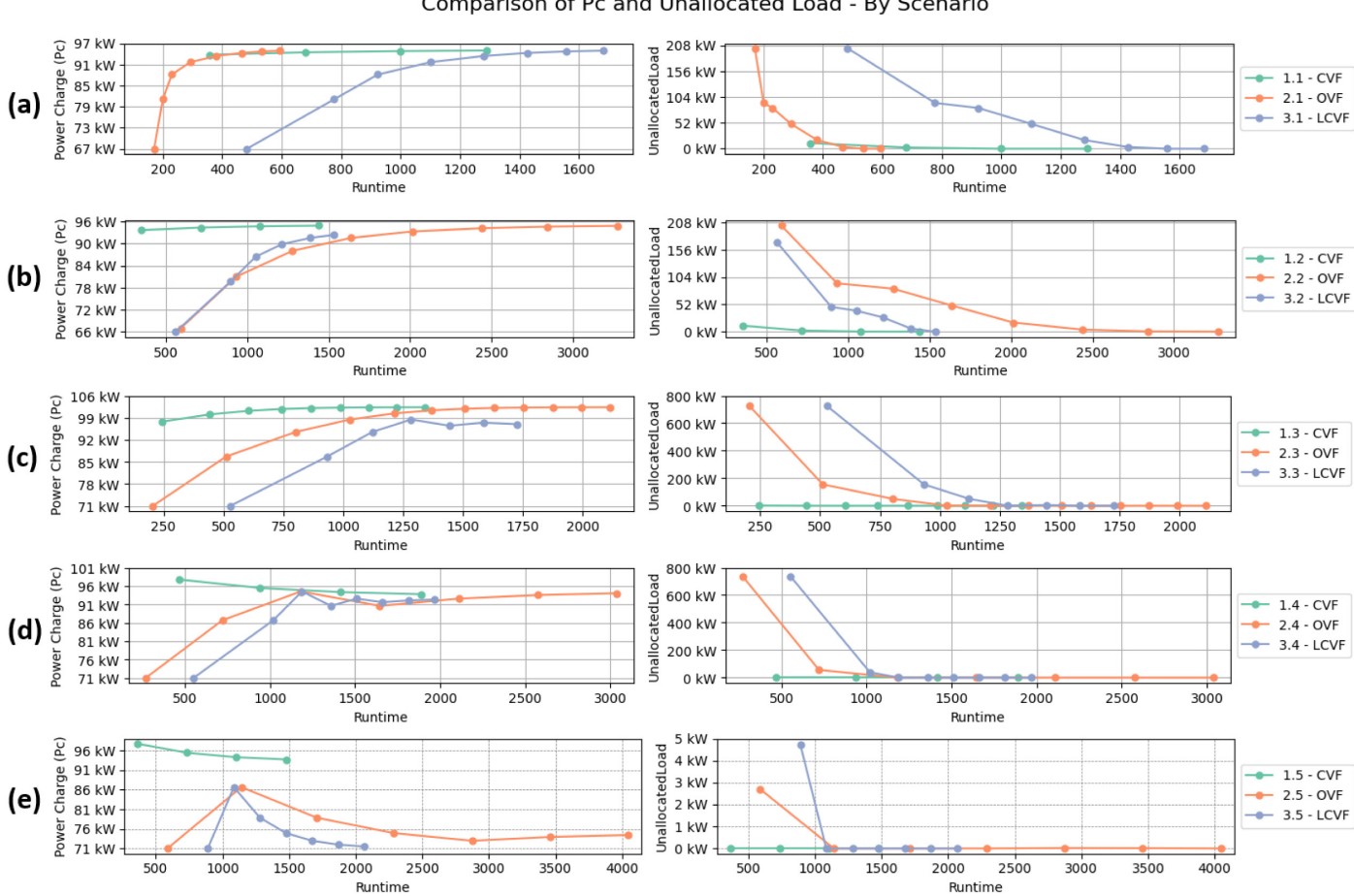

**Fig 13. Comparison of the results obtained in each scenario between the three techniques. (a)** 1st Use-Case: Original Scenario. **(b)** 2nd Use-Case: Flexible Scenario. **(c)** 3rd Use-Case: Increased Consumption. **(d)** 4th Use-Case: 8-Hour Availability. **(e)** 5th Use-Case: 24-Hour Availability.

In the first scenario (Fig 13(a)), it is observed that the OVF algorithm could accommodate all vehicle demands in approximately 600 seconds, in contrast to the CVF algorithm, which required around 1300 seconds, and the LCVF heuristic, which took over 1600 seconds to complete. This result was expected, as the OVF performs fewer operations in each iteration. Given the constraints of the first scenario (limited scheduling flexibility), there are many operations where no load is shifted from one iteration to the next. Consequently, CVF requires numerous initial operations, which extends its runtime, while LCVF has an even larger runtime due to the overhead associated with data recording in each iteration.

A comparative analysis of the Classic Valley-Filling (CVF) and Optimistic Valley-Filling (OVF) heuristics highlights differences in convergence speed and allocation efficiency, depending on different grid and charging conditions. CVF consistently showed faster convergence in scenarios with limited charging flexibility due to its conservative approach, which allocates EV loads quickly without complex iterative adjustments. Conversely, OVF, starting with a more optimistic initial charge value, was more efficient in high-flexibility scenarios, achieving an average reduction of up to 20% in the final $Pc$ value in high-availability contexts. This makes OVF advantageous in situations where charging demand can be distributed over a

broader time frame, enhancing load distribution efficiency and grid sustainability. Overall, CVF may be best suited for rapid charging demands, while OVF excels where greater temporal availability allows for more balanced allocations. These findings suggest that a hybrid approach, combining CVF's initial robustness with OVF's adaptability, could optimize performance across varied charging contexts.

In the original scenario, characterized by limited scheduling flexibility, the CVF heuristic showed a competitive advantage due to its fixed, conservative allocation strategy, which quickly stabilized EV loads. The OVF algorithm, in contrast, struggled with a lower initial load value, which limited its adjustments and led to a slower convergence rate. The OVF's optimistic initial setting proved less efficient here, as fewer adjustment opportunities exist when the time window is constrained. The LCVF heuristic performed steadily but required additional runtime due to the state storage mechanism, which, while enhancing stability, added to the runtime in a scenario with limited scheduling flexibility.

The scenario represented in Fig 13(b) indicates that electric vehicles have more power allocation opportunities before consuming electricity from the grid due to the slightly longer recharging time. The CVF technique presents faster convergence in this scenario, thus indicating that flexibility in recharging time allows the algorithm to determine the best time to recharge each vehicle. The runtime time of the OVF algorithm remains stable between iterations, indicating that even when no load allocation occurs, there are no performance benefits since the $Pc$ is reduced to a lower value, which is outside the power accommodation area. This stable time between iterations requires more time to achieve the convergence, thus satisfying the EV demand and penalizing the overall OVF runtime. On the other hand, LCVF had a very similar runtime to CVF, and in the first few iterations, LCVF could accommodate many vehicles by removing them from the recharge queue. As LCVF does not perform rework to the same vehicles in each iteration, the runtime trend between the iterations is reduced, improving its overall runtime.

In the flexible scenario, where EV charging demands were distributed across a broader time frame, the OVF algorithm excelled due to its lower initial charge value, allowing for finer adjustments over time and improving the load distribution. The LCVF heuristic also showed notable efficiency in stabilizing demand across iterations, as its state storage conserved previously allocated loads, reducing recalculations and fluctuations. The CVF algorithm maintained a stable performance; however, it was less adaptable to the benefits of increased flexibility, as it allocates based on a fixed peak demand criterion, resulting in less optimization than OVF and LCVF.

For the third use-case scenario, one may observe that by increasing the consumption factor of EVs, the heuristics begin to show significant differences in their results. Fig 13(c) shows that the LCVF algorithm achieved a lower $Pc$ (around 5%) compared to CVF at the cost of some impact on the runtime (see that LCVF finishes at time about 1750 seconds and CVF at time of 1350 seconds). The OVF algorithm had the worst execution time, and there was no significant improvement in the $Pc$ value in this scenario. The OVF heuristic had a near-constant time between iterations. The first iterations had a very low $Pc$, thus wasting time allocating the EV loads. The algorithm then continuously increases the $Pc$ to a level with a $Pc$ convergence.

The increased consumption scenario posed a challenging environment where the demand significantly strained each heuristic's ability to balance load effectively. The LCVF's state storage mechanism was particularly beneficial, as it retained allocations, reducing redundant processing in each iteration and yielding a 5% improvement in the $Pc$ parameter compared to CVF. OVF, however, was less efficient in this scenario due to the frequent need for iteration increases to meet the new demands, resulting in a longer runtime. The CVF heuristic maintained steady performance but lacked the adaptability to allocate in such high-demand scenarios without frequent recalculations, as required by OVF and LCVF.

The availability of TS for electric vehicles to connect to the electrical grid is increased in the last two remaining scenarios. In the fourth scenario (Fig 13(d)), each vehicle has at least 8 hours to recharge, while in the fifth scenario (Fig 13(e)), every vehicle can recharge at any time within 24 hours. This increase in the availability to charge on the network causes the CVF algorithm to introduce oscillatory behaviors, as described in Section 3.1.

The maximum $Pc$ varied slightly in the fourth scenario among the three heuristics (Fig 13 (d)). This quite similar $Pc$ final result is justified because the electrical energy peaks are present in the last 8-hour interval and, consequently, at the moment of greatest effort on the grid (Fig 11). Even when applying the valley-filling technique to stagger the load requested by electric vehicles, the time intervals in which allocations can occur end up being limited by the limit of 8 hours. This means that even when the lowest energy consumption (valley) occurs in this 8-hour period, its value is still close to the maximum energy demand (peak).

In the 8-hour availability scenario, where EVs had more connection time, OVF demonstrated enhanced adaptability in gradually balancing the demand load, achieving a lower peak charge with incremental improvements over CVF. The LCVF heuristic showed strong performance as well, as its state storage smoothed demand fluctuations and maintained consistent stability across iterations. This stability was less achievable in CVF due to its conservative approach, which resulted in higher peak charges during periods of increased flexibility. This scenario highlights the advantage of both OVF and LCVF in scenarios with moderate charging flexibility.

The fifth scenario (Fig 13(e)) presented the most significant differences in $Pc$ when comparing LCVF to the other two approaches. LCVF reduced approximately 25% in the $Pc$ value compared to CVF. This reduction is mostly due to allocating the EVs demands based on the lower $Pc$ value. Scaling with lower levels of $Pc$ ensures that no time-slot receives a significantly higher amount of demand than the others, balancing the electrical grid constantly and avoiding fluctuations. In this scenario, the OVF algorithm presented the worst performance, taking longer to reach the lowest $Pc$. One may also notice that LCVF outperformed the OVF approach considering the runtime. The load conservation strategy was why the algorithm had a faster convergence compared to OVF.

The 24-hour availability scenario provided a maximized scheduling window, allowing the LCVF heuristic to significantly outperform CVF and OVF by reducing the $Pc$ by up to 25%. This reduction is due to LCVF's load conservation approach, which minimizes the allocation load fluctuations, thus balancing demand and lowering peak requirements. OVF also benefited from this extended timeframe, though it required a longer runtime to achieve stability, as its optimistic starting value needed additional iterations to adapt to the longer scheduling window. The CVF algorithm, meanwhile, introduced oscillations that reduced its efficiency, showcasing LCVF's clear advantage in scenarios where increased availability demands a smoother allocation approach.

All heuristics met the electric vehicle demands for all scenarios. The graphs on the right in Fig 13 illustrates the power demand of the EVs according to the iterations of the algorithms. We highlight that both the OVF and the LCVF heuristics initialize the $Pc$ with a lower value than the CVF heuristic. In other words, LCVF and OVF do not present a faster convergence in the first interactions, but they may achieve better opportunities to meet the vehicle's charge demands with a lower $Pc$ than the CVF.

The graphical data in Fig 13 confirm each heuristic's capacity to handle different demand conditions, directly supporting the textual analysis. The runtime differences depicted across scenarios highlight OVF's relative efficiency under low-demand conditions and LCVF's advantages in scenarios with extended charging windows. The decreasing unallocated load curves across subfigures illustrate each heuristic's ability to meet EV demands while

highlighting how runtime performance varies with demand flexibility and heuristic design. These visual confirmations offer a clear comparison of efficiency among CVF, OVF, and LCVF under the studied conditions.

The runtime differences observed across heuristics in each scenario are shaped by both the complexity of the algorithms and the dataset size. In the CVF approach, the algorithm's fixed allocation strategy allows for faster execution in scenarios with limited scheduling flexibility, as it stabilizes quickly without extensive recalculations. However, with larger datasets, iterative recalculations increase exponentially, leading to slower performance. In contrast, OVF's optimistic initial setting improves adaptability in flexible scheduling scenarios but increases runtime in more constrained contexts, as additional iterations are needed to meet demand effectively. The LCVF heuristic, which conserves allocated loads across iterations, demonstrates notable efficiency with larger datasets, as it minimizes redundant computations and avoids reallocation. These runtime variations across algorithms and scenarios underscore the influence of problem complexity and dataset size on performance, with each heuristic demonstrating specific strengths depending on demand flexibility and dataset scale.

## 6.2 Statistical tests and validation

We have performed statistical tests comparing the OVF and LCVF heuristics to the CVF approach to validate the experiments and quantify the results. Statistical tests were performed considering the value of the load allocation delimiter $Pc$ and the total runtime of the algorithms. The experiments with each heuristic were run up to all EVs charging demands have been allocated so that the *UnallocatedLoad* value converged to zero. For all statistical tests, the significance level was set to $\alpha = 0.05$ to determine the levels of acceptance or rejection of the statistical tests in the p-value parameter.

The Shapiro-Wilk test [28] was conducted to determine if the $Pc$ values followed a normal distribution (Table 2). The test results indicated that the $Pc$ values did not follow a normal distribution, as evidenced by a p-value less than the significance level (p-value $<\alpha$). This finding is important, as many parametric statistical analyses, such as variance analysis, assume normality. Consequently, non-parametric tests were employed to ensure reliable data analysis. The non-normal distribution of EV load data is attributed to skewness in load demands, primarily

**Table 2. Summary of Shapiro-Wilk—*Pc* results.**

| Scenario | Heuristic | p-value | Normal Distribution |
|---|---|---|---|
| Original Scenario | CVF | 0.5381 | True |
|  | OVF | 0.0094 | False |
|  | LCVF | 0.0094 | False |
| Flexible Scenario | CVF | 0.5381 | True |
|  | OVF | 0.0094 | False |
|  | LCVF | 0.1151 | True |
| Increased Consumption | CVF | 0.0025 | False |
|  | OVF | 0.0002 | False |
|  | LCVF | 0.0071 | False |
| 8-Hour Availability | CVF | 0.5381 | True |
|  | OVF | 0.0048 | False |
|  | LCVF | 0.0010 | False |
| 24-Hour Availability | CVF | 0.5381 | True |
|  | OVF | 0.0845 | True |
|  | LCVF | 0.0450 | False |

**Table 3. Summary of Kruskal-Wallis—*Pc* results.**

| Scenario | p-value | Significant Difference |
|---|---|---|
| Original Scenario | 0.1659 | False |
| Flexible Scenario | 0.0254 | True |
| Increased Consumption | 0.0091 | True |
| 8-Hour Availability | 0.0387 | True |
| 24-Hour Availability | 0.0108 | True |

due to peak usage times and varying charging requirements among EVs, creating an asymmetrical distribution that necessitates non-parametric tests for accurate analysis.

The non-parametric Kruskal-Wallis test [29] was applied to the *Pc* to identify statistically significant differences among the results (Table 3). Upon performing the test, we discovered at least one significant difference among the results of each technique for all proposed scenarios (p-values $<\alpha$).

Once significant differences have been identified, the subsequent step involves precisely identifying where these differences occur. To carry out this task, we used a post-hoc test: the Dunn test [30] that compares the heuristics within each scenario to identify the specific cases where such differences manifest themselves. The hypotheses formulated for considering that the results of the heuristics can be statistically different are as follows:

$$\mathbf{H_0}: \quad \text{No significant difference between the } Pc \text{ values means.}$$

$$\mathbf{H_1}: \quad \text{Significant difference between the } Pc \text{ values means.}$$

The null hypothesis ($H_0$) assumes no significant difference between the compared means, while the alternative hypothesis ($H_1$) indicates significant performance differences between techniques. The test compares all possible combinations of techniques, identifying which pairs present significant differences, allowing us to conclude whether these differences are statistically relevant or attributable by chance.

The Dunn test summary, shown in Table 4, compares each heuristic pair and includes columns for the scenario, heuristic pairs, mean difference, adjusted p-value, and "Reject

**Table 4. Summary of Dunn test—*Pc* results.**

| Scenario | Comparação | p-adj | Reject $H_0$ |
|---|---|---|---|
| Original Scenario | CVF—OVF | 0.2507 | False |
| | CVF—LCVF | 0.2507 | False |
| | OVF—LCVF | 1.0000 | False |
| Flexible Scenario | CVF—OVF | 0.3004 | False |
| | CVF—LCVF | 0.0203 | True |
| | OVF—LCVF | 0.5093 | False |
| Increased Consumption | CVF—OVF | 1.0000 | False |
| | CVF—LCVF | 0.0089 | True |
| | OVF—LCVF | 0.0535 | False |
| 8-Hour Availability | CVF—OVF | 0.1241 | False |
| | CVF—LCVF | 0.0383 | True |
| | OVF—LCVF | 1.0000 | False |
| 24-Hour Availability | CVF—OVF | 0.0347 | True |
| | CVF—LCVF | 0.0124 | True |
| | OVF—LCVF | 1.0000 | False |

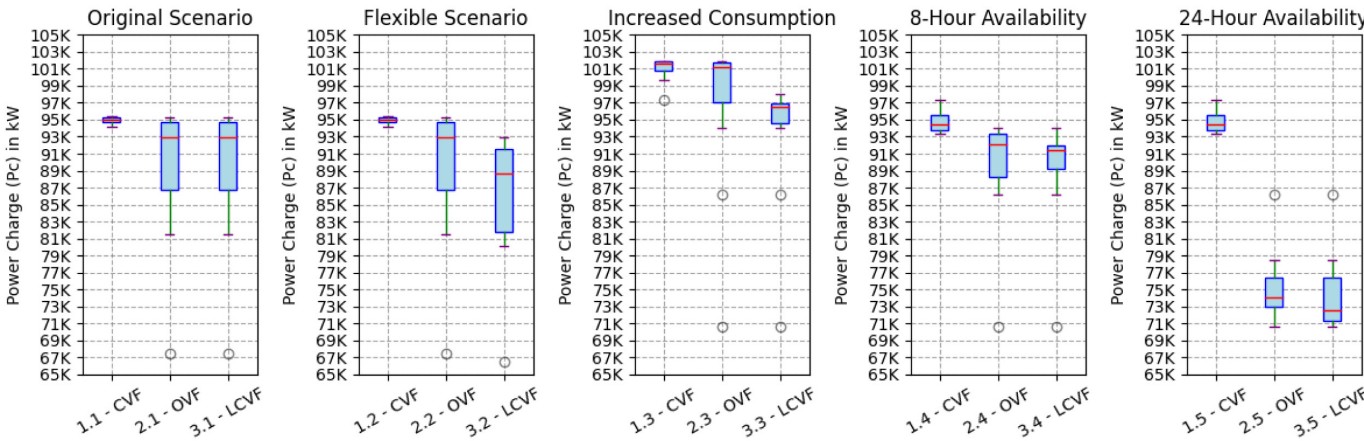

**Fig 14. Distribution of *Pc* values of the heuristics in each scenario.**

$H_0$", indicating whether $H_0$ (no significant difference) or $H_1$ (significant difference) is accepted.

Fig 14 outlines the dispersion of the *Pc* values from each heuristic and scenario. The outliers values (blank circles) in this figure represent the initial points of the *Pc* value, which, given the heuristic configuration, employs the binary search method for convergence. The binary search method is an efficient algorithm for finding a target value within a sorted array by repeatedly dividing the search interval in half and narrowing down the possible locations of the target until convergence is achieved. The scenario with greater availability (24-Hour Availability) shows a greater dispersion between CVF and the proposed heuristics, thus corroborating the significant differences informed in Table 4. This means that the greater the availability of the electric vehicle, the greater its allocation capacity will be, considerably mitigating the effort exerted on the electrical grid.

The analysis of power charge scenarios using CVF, OVF, and LCVF techniques reveals distinct responses under varying consumption and charging availability conditions. OVF adapts effectively in flexible scenarios with limited recharging windows but exhibits greater sensitivity to peak demand when consumption increases. LCVF is more efficient at reducing load under extended availability (24 hours), promoting a balanced energy demand distribution. CVF, in contrast, remains consistent and stable across all scenarios. Outliers, especially in the Flexible and 24-Hour Availability scenarios, demonstrate the flexibility of OVF and LCVF in managing extreme variations, achieving significant load reductions during low-demand periods (e.g., below 67 kW). This adaptability supports strategic load distribution, making OVF and LCVF ideal choices when the goal is to maximize efficiency by distributing the load within flexible timeslots.

We used the accumulated runtime throughout the iterations to verify whether the heuristics had significant differences in their runtimes. This accumulated runtime reflects the general performance trend throughout the process, highlighting points where significant differences arise in electric vehicles' convergence and load distribution. Following the same statistical analysis procedure of the *Pc* values, we first apply the Shapiro-Wilk test to observe if the runtime distribution follows a normal distribution (Table 5). The total runtime for each heuristic presented a normal distribution for all experiments (p-value $>\alpha$).

**Table 5. Summary of Shapiro-Wilk—Runtime results.**

| Scenario | Heuristic | p-value | Normal Distribution |
|---|---|---|---|
| Original Scenario | CVF | 0.9621 | True |
| | OVF | 0.4918 | True |
| | LCVF | 0.9076 | True |
| Flexible Scenario | CVF | 0.9707 | True |
| | OVF | 0.9190 | True |
| | LCVF | 0.9472 | True |
| Increased Consumption | CVF | 0.9404 | True |
| | OVF | 0.6990 | True |
| | LCVF | 0.8688 | True |
| 8-Hour Availability | CVF | 0.9706 | True |
| | OVF | 0.9449 | True |
| | LCVF | 0.8963 | True |
| 24-Hour Availability | CVF | 0.9719 | True |
| | OVF | 0.9435 | True |
| | LCVF | 0.9492 | True |

The analysis of variance tests (ANOVA) [31] indicate significant differences among the heuristics' runtimes. ANOVA was chosen to test for differences in group means, under the assumptions of normality, homogeneity of variances, and independence of observations, verified as follows:

- **Data Normality:** Verified using the Shapiro-Wilk test, confirming that each group's data followed a normal distribution.

- **Homogeneity of Variances:** Assessed with Levene's test, which indicated comparable variances across groups.

- **Independence of Observations:** Ensured by collecting runtime measurements independently for each scenario.

In each scenario, the p-value of the ANOVA test is lower than the significance level (p-value $<\alpha$), indicating a statistical difference in runtime (Table 6). This result suggests that variations across scenarios impact algorithm execution times, which is critical for interpreting results and selecting suitable configurations. Specifically, only the Original Scenario exhibits a statistically significant difference, suggesting it has a unique impact on algorithm runtime. This insight is valuable for selecting configurations, as the Original Scenario may require specific computational adjustments. Understanding these differences aids in making informed decisions regarding heuristic approaches. For borderline p-values, additional methods such as

**Table 6. Summary ANOVA—Runtime results.**

| Scenario | p-value | Significant Difference |
|---|---|---|
| Original Scenario | 0.0006 | True |
| Flexible Scenario | 0.0622 | False |
| Increased Consumption | 0.0854 | False |
| 8-Hour Availability | 0.5798 | False |
| 24-Hour Availability | 0.0538 | False |

**Table 7. Summary of Tukey HSD test—Runtime results.**

| Scenario | Comparison | p-adj | Reject $H_0$ |
|---|---|---|---|
| Original Scenario | CVF—OVF | 0.0766 | False |
| | CVF—LCVF | 0.2737 | False |
| | OVF—LCVF | 0.0004 | True |
| Flexible Scenario | CVF—OVF | 0.0916 | False |
| | CVF—LCVF | 0.8888 | False |
| | OVF—LCVF | 0.1419 | False |
| Increased Consumption | CVF—OVF | 0.0781 | False |
| | CVF—LCVF | 0.2724 | False |
| | OVF—LCVF | 0.9014 | False |
| 8-Hour Availability | CVF—OVF | 0.5718 | False |
| | CVF—LCVF | 0.8908 | False |
| | OVF—LCVF | 0.7677 | False |
| 24-Hour Availability | CVF—OVF | 0.0537 | False |
| | CVF—LCVF | 0.5674 | False |
| | OVF—LCVF | 0.2069 | False |

post hoc tests or data transformations may be useful to confirm findings or address assumption violations.

The comparative analysis of the runtime between each pair of heuristics (Tukey HSD test [32]) in each scenario is presented in Table 7. The Original scenario was the only one in which the post-hoc test (Tukey HSD) indicated a statistically significant difference. In this scenario, the OVF runtime was significantly lower than LCVF. OVF has a near-constant runtime in each iteration, while LCVF requires larger intervals in the first iterations. There is little flexibility to carry out the necessary EV power charging allocation in the first scenario, so that the accommodation of EV power demands occur mainly with higher $Pc$ values, that is, in the last iterations, which penalizes the LCVF heuristic.

Likewise, we can observe the differences among the heuristics runtimes through the boxplot graph presented in Fig 15. The graph shows a difference in the Original Scenario when

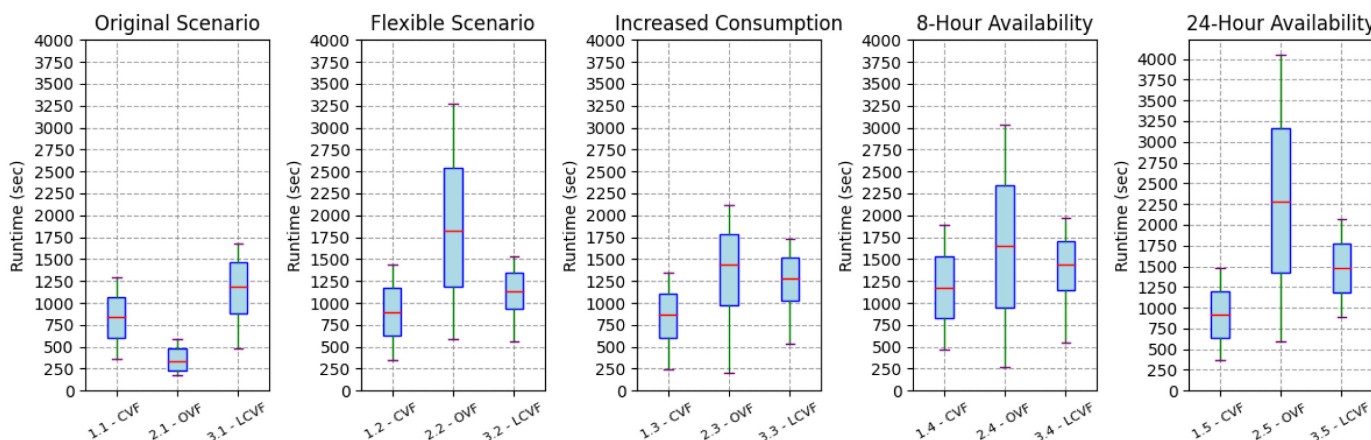

**Fig 15. Distribution of runtime of the heuristics in each scenario.**

comparing the OVF and LCVF heuristics. In the 24-Hour Availability scenario, although the Tukey HSD test, presented in Table 7, does not indicate any significant difference in the pairwise tests, the p-value of the comparison between CVF and OVF is quite close to the rejection threshold (p-value = 0.0537, while we reject the null hypothesis when p-value $< \alpha$).

As shown in Fig 15, the runtime analysis reveals that the OVF technique tends to have longer runtimes in scenarios requiring greater flexibility, such as the Flexible and 24-Hour Availability scenarios, with median runtimes ranging from 1500 to 2000 seconds and peaks near 3750 seconds, reflecting the increased computational demands of load adjustments. The LCVF technique shows higher runtimes in scenarios with restricted charging windows, notably in the 8-Hour Availability scenario, where the median reaches approximately 1500 seconds, indicating a higher processing requirement for load balancing within shorter timeframes. In contrast, the CVF maintains consistent runtimes with medians around 1000 seconds across all scenarios, although it generally requires more time on average than OVF and LCVF in scenarios with less flexibility, such as the Original Scenario. These results suggest that the optimal technique selection depends on balancing processing time with the need for adaptability within the available charging window.

The statistical analysis validated the load distribution of electric vehicles in different scenarios using each heuristic. The tests allowed us to indicate statistically significant differences in $Pc$ values and algorithm runtimes. Specifically, notable disparities between the CVF and LCVF algorithms stand out, showing that the latter outperforms the classical approach for all scenarios and presents a substantially different load allocation in different situations, especially in the 24-hour availability scenario.

The analysis of runtimes also revealed different algorithms' sensitivities in different contexts, underscoring the practical importance of considering such variations when interpreting the performance results for a specific use-case scenario. These results indicate that scenario variations influence algorithm runtimes differently, significantly affecting the choice of the most appropriate technique for different application contexts.

To evaluate the real performance of the OVF and LCVF, we analyze the $Pc$ values from each algorithm and the impact of these values on the accommodation of EVs for recharging. The comparison data is presented in Table 8. The table compares CVF to OVF and LCVF. We compared the average $Pc$ values and the last $Pc$ from each algorithm, calculating their difference. The differences between the absolute values of $Pc$ (in kW) are determined as follows:

$$\text{Differences (kW)} = Pc_{\text{OVF|LCVF}} - Pc_{\text{CVF}} \tag{3}$$

**Table 8. Comparative results (in kW) of scenarios and heuristics.**

| Scenarios | CVF Pc Data | | Heuristics Pc Data | | | Differences | | | |
| | Mean Pc (kW) | Last Pc (kW) | Heuristics | Mean Pc (kW) | Last Pc (kW) | Mean Pc (kW) | Mean Pc (%) | Last Pc (kW) | Last Pc(%) |
|---|---|---|---|---|---|---|---|---|---|
| **Original Scenario** | 94,885.67 | 95,366.85 | OVF | 88,531.52 | 95,321.89 | -6,354.15 | -6.70 | -44.96 | -0.05 |
| | | | LCVF | 88,531.52 | 95,321.89 | -6,354.15 | -6.70 | -44.96 | -0.05 |
| **Flexible Scenario** | 94,885.67 | 95,366.85 | OVF | 88,531.52 | 95,321.89 | -6,354.15 | -6.70 | -44.96 | -0.05 |
| | | | LCVF | 84,783.48 | 92,870.73 | -10,102.20 | -10.65 | -2,496.12 | -2.62 |
| **Increased Consumption** | 100,890.65 | 101,889.07 | OVF | 96,685.95 | 101,891.67 | -4,204.70 | -4.17 | 2.60 | 0.00 |
| | | | LCVF | 91,205.93 | 96,521.48 | -9,684.72 | -9.60 | -5,367.59 | -5.27 |
| **8-Hour Availability** | 94,890.21 | 93,315.02 | OVF | 88,548.16 | 93,583.94 | -6,342.06 | -6.68 | 268.92 | 0.29 |
| | | | LCVF | 88,473.85 | 91,870.38 | -6,416.37 | -6.76 | -1,444.64 | -1.55 |
| **24-Hour Availability** | 94,890.21 | 93,315.02 | OVF | 75,678.93 | 74,000.34 | -19,211.28 | -20.25 | -19,314.68 | -20.70 |
| | | | LCVF | 74,979.52 | 71,062.80 | -19,910.69 | -20.98 | -22,252.22 | -23.85 |

The percentage difference was also calculated for each combination of scenario and heuristic, and it compares the values resulting from OVF and LCVF to the CVF (in %). Evaluating the differences between each pair of algorithms in each scenario provides a clear understanding of the benefits and limitations of the approaches. The formula used to calculate the percentage difference is:

$$\text{Differences } (\%) = \left( \frac{\text{Pc}_{\text{OVF|LCVF}} - \text{Pc}_{\text{CVF}}}{\text{Pc}_{\text{CVF}}} \right) \times 100\% \tag{4}$$

Most of the mean difference values are negative, suggesting that the $Pc$ mean values of the second algorithm are greater than the first. As an example, in the Original Scenario, the mean difference between CVF and OVF was -6,354.15, meaning that the mean $Pc$ values of the OVF algorithm were 6,354.15 kW greater than the mean $Pc$ value of the CVF algorithm. The results of the Mean Difference column show that our proposed algorithms outperform the classical valley-filling algorithm for all scenarios. Despite that, the results from the posthoc test (Table 8) indicate no statistically significant differences in the $Pc$ values among the algorithms in each scenario. Only in the fifth scenario (24-hour availability) did CVF show significant differences between the OVF and LCVF algorithms, with an average difference in $Pc$ values of roughly 20 kW. The OVF and LCVF algorithms did not present significant differences in this scenario.

One may observe that most of the mean difference values are negative, suggesting that the $Pc$ mean values of the second algorithm are greater than the first. As an example, in the Original Scenario, the mean difference between CVF and OVF was -6,354.15, meaning that the mean $Pc$ values of the OVF algorithm were 6,354.15 kW greater than the mean $Pc$ value of the CVF algorithm. The results of the Mean Difference column show that our proposed algorithms outperform the classical valley-filling algorithm for all scenarios. Despite that, the results from the posthoc test (Table 8) indicate no statistically significant differences in the $Pc$ values among the algorithms in each scenario. Only in the fifth scenario (24-hour availability) did CVF show significant differences between the OVF and LCVF algorithms, with an average difference in $Pc$ values of roughly 20 kW. The OVF and LCVF algorithms did not present significant differences in this scenario.

The data reveals the performance of the algorithms in various scenarios, measuring the impact on the average and final values of $Pc$ (number of parts) and their respective percentage differences and energy consumption in kW. In the original scenario, both OVF and LCVF algorithms showed a similar decrease (6.70%) in the mean $Pc$, meaning that, on average, the $Pc$ of these algorithms had a lower value than CVF along the iterations. This decrease corresponds to 6,354.15 kW in average consumption. However, in the final result (Last $Pc$), the decrease was just 0.05%, meaning that the CVF algorithm converged to the same $Pc$ values as OVF and LCVF.

In the flexible scenario, the OVF maintained a 6.70% reduction in the mean value of $Pc$ (6,354.15 kW), while the LCVF presented a more significant benefit of 10.65% (10,102.20 kW) and 2.62% in the final value of $Pc$ (2,496.12 kW). In the increased consumption scenario, the OVF heuristic resulted in a more moderate decrease of 4.17% in the mean $Pc$ (4,204.70 kW), with an insignificant difference in the final $Pc$ value, while the LCVF showed a reduction of 9.60% in the average value of $Pc$ (9,684.72 kW) and 5.27% in the final value of $Pc$ (5,367.59 kW).

The 8-hour and 24-hour availability scenarios significantly impacted the performance of the OVF and LCVF heuristics. In the 8-hour scenario, the Mean $Pc$ values achieved decreases of 6.68% (OVF: 6,342.06 kW) and 6.76% (LCVF: 6,416.37 kW), with slight variations in final

*Pc* values. In the 24-hour scenario, there was a drastic reduction in mean *Pc* values, with OVF 20.25% (19,211.28 kW) and LCVF 20.98% (19,910.69 kW), and the final *Pc* up to 23.85% (22,252.22 kW) for LCVF. These results underscore the adaptability and effectiveness of the OVF and LCVF heuristics in scenarios with different availability durations.

## 6.3 Synthesis of results and practical implications

The analysis revealed that the LCVF heuristic achieved an average convergence rate improvement of 15% over CVF and an 8% reduction in resource usage compared to OVF. These reductions in $P_c$ highlight LCVF's effectiveness in balancing resource allocation, particularly in high-demand scenarios. While LCVF excels in peak reduction and load stability, CVF and OVF each offer specific advantages. In scenarios with volatile demand, CVF adapts quickly, minimizing peak deviations by up to 7%. OVF, in contrast, performs well with steady demand, achieving a load factor improvement of approximately 5% without frequent adjustments.

A more detailed analysis of these heuristics could guide strategic selection based on scenario characteristics. For instance, CVF may be preferred for real-time applications with short-term demand fluctuations, while OVF could be ideal for scenarios with stable, predictable loads. Evaluating these trade-offs across varied datasets would allow future research to optimize heuristic selection to align with specific grid conditions, ultimately enhancing system resilience.

The analysis of scenarios with increased flexibility and consumption revealed statistically significant differences between the heuristics; however, the practical relevance of these differences requires careful consideration to guide decision-making. In flexible charging scenarios, redistributing load to off-peak periods proved advantageous, allowing cost savings by lowering peak demand requirements and avoiding grid overload. The LCVF heuristic demonstrated practical benefits here, achieving a reduction of up to 10.65% in average peak load (*Pc*) compared to the CVF heuristic, representing substantial energy resource savings. This redistribution capacity of the LCVF heuristic enhances grid stability, making it a suitable choice for systems with high electric vehicle (EV) adoption where load-balancing efficiency is crucial.

In contrast, in high-consumption scenarios, small differences in the peak load reduction between heuristics, such as OVF's 4.17% reduction versus LCVF's 9.60%, indicate that while both approaches are viable, their practical benefits may vary based on specific system demands and cost-effectiveness. For example, although OVF showed moderate peak reduction, the added computational cost of LCVF might be justified in high-demand environments where optimal resource allocation is essential. Consequently, the decision to select a heuristic should factor in both statistical significance and practical impact, particularly in high-demand urban grids where minimizing peak loads directly contributes to grid resilience and reduces the need for costly infrastructure upgrades. This dual consideration provides a robust foundation for selecting load allocation strategies in EV charging systems.

While the LCVF heuristic has demonstrated significant improvements in peak load reduction and grid stability, its implementation presents computational limitations. The iterative nature of the LCVF and the need for continuous updates to load allocations contribute to increased computational complexity, particularly as the number of EVs and time slots increase. In large-scale networks, such as city-wide grids, this complexity may exceed available processing capabilities, posing challenges for real-time applications. Optimizations or parallel processing strategies could help address these challenges, enabling the application of the LCVF in broader scenarios.

These findings underscore the potential impact of the LCVF heuristic on real-world electric vehicle (EV) charging systems, particularly in urban settings with high EV adoption rates. By effectively reducing peak loads, the LCVF approach can help prevent grid overloads, enable

smoother integration of renewable energy sources, and facilitate cost reductions through optimized energy distribution. Integrating the LCVF heuristic into current EV infrastructures could significantly enhance grid stability, making it feasible to handle increased EV charging demands without extensive grid upgrades.

Future research could explore the scalability of the LCVF heuristic across diverse grid configurations and under varying demand scenarios. Additionally, studies incorporating real-time data from EV charging stations and renewable energy sources would be valuable in assessing the heuristic's adaptability and long-term effectiveness. Analyzing the LCVF's performance alongside emerging smart grid technologies may yield further insights, supporting the development of comprehensive energy management strategies.

**6.3.1 Challenges in real-world LCVF implementation.** Implementing the proposed VF heuristics in real-world scenarios presents several challenges. A significant obstacle is the computational capacity required to perform iterative load allocations efficiently, particularly in large-scale power grids with high numbers of electric vehicles (EVs) and fluctuating power demands. Such systems may encounter computational constraints that impede real-time scheduling, making optimizations or simplified approaches necessary for practical application. Additionally, user behavior introduces unpredictability, as charging times, energy requirements, and vehicle availability vary widely, impacting the scheduling accuracy and effectiveness of VF strategies. Furthermore, unique grid characteristics, such as existing load profiles, peak demand periods, and network distribution constraints, could affect the efficacy of VF approaches. Addressing these challenges demands flexible algorithms capable of adapting to real-world variability while maintaining performance efficiency. Future studies could explore adaptive VF models that account for user unpredictability and computational limitations.

**6.3.2 Potential applications of LCVF approach.** Beyond electric vehicle (EV) charging, the VF approach offers potential applications across various sectors that require load balancing to enhance efficiency and stability. For instance, VF can be applied in renewable energy integration, where periods of high renewable generation (e.g., solar or wind) align with lower demand periods, filling valleys and stabilizing supply without straining grid resources. Additionally, VF techniques could improve energy storage management by optimizing charging schedules for large-scale battery systems, ensuring charging occurs during off-peak hours to bolster grid resilience and reduce operational costs. Another potential application is in data centers, where VF can manage power-intensive tasks by scheduling them during low-demand periods, ultimately reducing peak loads and energy expenses. The versatility of VF underscores its broader relevance across industries that prioritize energy efficiency and grid stability.

# 7 Conclusions

This paper introduces a new heuristic, termed Load Conservation Valley-Filling (LCVF), designed to manage power allocation demands from electric vehicles (EVs) within the electrical grid. LCVF builds upon the Classical Valley-Filling (CVF) method by incorporating an optimistic approach for calculating the *Pc* parameter and employing memoization to retain charging load allocations across iterations. This memoization facilitates a more efficient distribution of load over time, reducing the need for frequent adjustments. Furthermore, the optimistic calculation of the *Pc* parameter allows LCVF to allocate load with higher precision during high-demand periods, minimizing energy wastage compared to traditional methods.

The analysis of the results highlights the distinct advantages of the LCVF algorithm over other approaches, such as CVF and OVF, in managing electric vehicle load. In scenarios like Original and Flexible, LCVF achieved a significant reduction in peak charge (*Pc*), with a 10.65% decrease compared to CVF, equating to an absolute reduction of 10,102.20 kW. In the

Increased Consumption scenario, LCVF provided even greater benefits, achieving a 9.60% reduction compared to CVF, equivalent to 9,684.72 kW.

LCVF maintained consistently lower average energy consumption across various scenarios. For instance, in the 8-Hour Availability scenario, this heuristic achieved a 6.76% reduction compared to CVF, representing 6,416.37 kW. In the 24-Hour Availability scenario, LCVF was even more effective, showing a 20.98% reduction in average energy consumption relative to CVF, corresponding to 19,910.69 kW, thus demonstrating its effectiveness in managing load over extended periods.

These quantitative reductions not only improve energy efficiency but also have substantial practical implications, such as cost savings for grid operators, who benefit from reduced energy demands during peak hours. Additionally, the reduced load during these periods contributes to grid stability, which is particularly beneficial in distribution networks with high renewable penetration. In commercial applications, these advantages could translate into significant annual savings and decrease the need for additional investment in generation infrastructure.

These findings reinforce the benefits of the LCVF heuristic in enhancing energy efficiency and optimizing EV load distribution. LCVF's ability to maintain consistently lower energy consumption across scenarios, coupled with its rapid convergence to *near-optimal* load values, positions it as a robust and effective solution for intelligent EV load management, contributing to the sustainability and efficiency of electrical energy systems.

Compared to the classical approach, the OVF heuristic has limitations in scenarios where the power required to allocate all EV demands approaches the upper bound (*upperPc*). In such cases, OVF may require more iterations to achieve convergence of the *Pc* constraint and may be less ideal when demand forecasting is uncertain, as it depends on accurately identifying low-demand periods. While LCVF is more efficient in conserving load and iterating on previous results for continuous optimization, its computational complexity can increase in scenarios with varied demand data, potentially extending runtime and reducing feasibility for real-time applications. To address these challenges, parallel processing and memory optimization techniques could reduce processing time. Alternatively, data subsampling for less critical scenarios may provide a practical solution, maintaining accuracy while enhancing efficiency.

The LCVF heuristic offers practical advantages for energy sector professionals, including grid distribution managers and EV station operators. By aligning charging schedules with low-demand periods, LCVF helps maximize renewable energy integration while reducing reliance on non-renewable sources during peak hours. This capability is particularly relevant for operators seeking to balance grid stability with rising energy demands from widespread EV adoption.

Future work could explore integrating smart charging technologies and vehicle-to-grid (V2G) strategies to further enhance grid efficiency and stability. Additional research could also focus on machine learning techniques to leverage historical EV charging data, enabling allocation predictions for extended periods, such as a week or month in advance. Accurate weekly EV demand forecasts could significantly benefit grid operators in power supply planning. Real-time optimization and dynamic adaptation of the LCVF algorithm to accommodate constantly changing grid conditions represent another promising research avenue. Furthermore, analyzing the economic and environmental impact of *large-scale* heuristic adoption could provide valuable insights for utility companies interested in demand management policies and sustainable development in the context of EVs.

Among potential directions, integrating machine learning models like recurrent neural networks (RNNs) and sequential prediction methods could enhance LCVF's ability to anticipate load demands. For real-time optimization, Model Predictive Control (MPC) algorithms could

support dynamic adjustments to grid conditions, maximizing allocation efficiency over time. These advancements would strengthen LCVF's responsiveness to sudden changes, making it more viable for large-scale applications.

## Author Contributions

**Conceptualization:** Guilherme Gloriano de Souza, Ricardo Ribeiro dos Santos.

**Data curation:** Guilherme Gloriano de Souza, Ricardo Ribeiro dos Santos.

**Formal analysis:** Guilherme Gloriano de Souza, Ricardo Ribeiro dos Santos, Ruben Barros Godoy.

**Funding acquisition:** Ricardo Ribeiro dos Santos.

**Investigation:** Guilherme Gloriano de Souza.

**Methodology:** Guilherme Gloriano de Souza, Ricardo Ribeiro dos Santos.

**Supervision:** Ricardo Ribeiro dos Santos.

**Validation:** Guilherme Gloriano de Souza, Ruben Barros Godoy.

**Visualization:** Guilherme Gloriano de Souza.

**Writing – original draft:** Guilherme Gloriano de Souza, Ricardo Ribeiro dos Santos.

**Writing – review & editing:** Guilherme Gloriano de Souza, Ricardo Ribeiro dos Santos, Ruben Barros Godoy.

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
