## [Decision Letter · Decision Letter 0]

27 Sep 2024

PONE-D-24-35622A load conservation valley-filling heuristic for electric vehicle power charging allocation: algorithms and scenariosPLOS ONE

Dear Dr. Souza,

Thank you for submitting your manuscript to PLOS ONE. After careful consideration, we feel that it has merit but does not fully meet PLOS ONE’s publication criteria as it currently stands. Therefore, we invite you to submit a revised version of the manuscript that addresses the points raised during the review process.

Please submit your revised manuscript by Nov 11 2024 11:59PM If you will need more time than this to complete your revisions, please reply to this message or contact the journal office at plosone@plos.org. Please include the following items when submitting your revised manuscript:A rebuttal letter that responds to each point raised by the academic editor and reviewer(s). You should upload this letter as a separate file labeled 'Response to Reviewers'.A marked-up copy of your manuscript that highlights changes made to the original version. You should upload this as a separate file labeled 'Revised Manuscript with Track Changes'.An unmarked version of your revised paper without tracked changes. You should upload this as a separate file labeled 'Manuscript'.

We look forward to receiving your revised manuscript.

Kind regards,

Barry Kweh

Academic Editor

PLOS ONE

Journal Requirements:

1. When submitting your revision, we need you to address these additional requirements. Please ensure that your manuscript meets PLOS ONE's style requirements, including those for file naming. The PLOS ONE style templates can be found at https://journals.plos.org/plosone/s/file?id=wjVg/PLOSOne_formatting_sample_main_body.pdf and https://journals.plos.org/plosone/s/file?id=ba62/PLOSOne_formatting_sample_title_authors_affiliations.pdf 2. We suggest you thoroughly copyedit your manuscript for language usage, spelling, and grammar. If you do not know anyone who can help you do this, you may wish to consider employing a professional scientific editing service.  The American Journal Experts (AJE) (https://www.aje.com/) is one such service that has extensive experience helping authors meet PLOS guidelines and can provide language editing, translation, manuscript formatting, and figure formatting to ensure your manuscript meets our submission guidelines. Please note that having the manuscript copyedited by AJE or any other editing services does not guarantee selection for peer review or acceptance for publication.  Upon resubmission, please provide the following: The name of the colleague or the details of the professional service that edited your manuscript A copy of your manuscript showing your changes by either highlighting them or using track changes (uploaded as a *supporting information* file) A clean copy of the edited manuscript (uploaded as the new *manuscript* file) 3. Please note that PLOS ONE has specific guidelines on code sharing for submissions in which author-generated code underpins the findings in the manuscript. In these cases, we expect all author-generated code to be made available without restrictions upon publication of the work. Please review our guidelines at https://journals.plos.org/plosone/s/materials-and-software-sharing#loc-sharing-code and ensure that your code is shared in a way that follows best practice and facilitates reproducibility and reuse. 4. Thank you for stating the following financial disclosure: "Conselho Nacional de Desenvolvimento Científico e Tecnológico Award Number: 160179/2020-3" Please state what role the funders took in the study.  If the funders had no role, please state: "The funders had no role in study design, data collection and analysis, decision to publish, or preparation of the manuscript." If this statement is not correct you must amend it as needed. Please include this amended Role of Funder statement in your cover letter; we will change the online submission form on your behalf. 5. Thank you for stating the following in the Acknowledgments Section of your manuscript: "The authors would like to thank the Conselho Nacional de Desenvolvimento Cient´ıfico e 809Tecnol´ogico (CNPq), Grant No.: 160179/2020-3, for their financial support to this work. They also extend their gratitude to the Universidade Federal de Mato Grosso do Sul (UFMS) and the Research Laboratory of High-Performance Computing Systems (LSCAD), where this work was conducted". We note that you have provided funding information that is not currently declared in your Funding Statement. However, funding information should not appear in the Acknowledgments section or other areas of your manuscript. We will only publish funding information present in the Funding Statement section of the online submission form. Please remove any funding-related text from the manuscript and let us know how you would like to update your Funding Statement. Currently, your Funding Statement reads as follows: "The authors would like to thank the Conselho Nacional de Desenvolvimento Cient´ıfico e 809Tecnol´ogico (CNPq), Grant No.: 160179/2020-3, for their financial support to this work. They also extend their gratitude to the Universidade Federal de Mato Grosso do Sul (UFMS) and the Research Laboratory of High-Performance Computing Systems (LSCAD), where this work was conducted". Please include your amended statements within your cover letter; we will change the online submission form on your behalf. 6. We note that your Data Availability Statement is currently as follows: "All relevant data are within the manuscript and its Supporting Information files." Please confirm at this time whether or not your submission contains all raw data required to replicate the results of your study. Authors must share the “minimal data set” for their submission. PLOS defines the minimal data set to consist of the data required to replicate all study findings reported in the article, as well as related metadata and methods (https://journals.plos.org/plosone/s/data-availability#loc-minimal-data-set-definition). For example, authors should submit the following data: - The values behind the means, standard deviations and other measures reported;- The values used to build graphs;- The points extracted from images for analysis. Authors do not need to submit their entire data set if only a portion of the data was used in the reported study. If your submission does not contain these data, please either upload them as Supporting Information files or deposit them to a stable, public repository and provide us with the relevant URLs, DOIs, or accession numbers. For a list of recommended repositories, please see https://journals.plos.org/plosone/s/recommended-repositories. If there are ethical or legal restrictions on sharing a de-identified data set, please explain them in detail (e.g., data contain potentially sensitive information, data are owned by a third-party organization, etc.) and who has imposed them (e.g., an ethics committee). Please also provide contact information for a data access committee, ethics committee, or other institutional body to which data requests may be sent. If data are owned by a third party, please indicate how others may request data access.

Additional Editor Comments:

A well written interesting article which requires greater structure, a broader discussion of the literature in both a tabulated as well as written format as suggested by reviewers and clarification of the methodology.

Reviewers' comments:

Reviewer's Responses to Questions

**Comments to the Author**

1. Is the manuscript technically sound, and do the data support the conclusions?

Reviewer #1: Yes

Reviewer #2: Yes

Reviewer #3: Partly

Reviewer #4: Yes

Reviewer #5: No

2. Has the statistical analysis been performed appropriately and rigorously? 

Reviewer #1: Yes

Reviewer #2: No

Reviewer #3: Yes

Reviewer #4: Yes

Reviewer #5: No

3. Have the authors made all data underlying the findings in their manuscript fully available?

Reviewer #1: Yes

Reviewer #2: Yes

Reviewer #3: Yes

Reviewer #4: Yes

Reviewer #5: No

4. Is the manuscript presented in an intelligible fashion and written in standard English?

Reviewer #1: Yes

Reviewer #2: Yes

Reviewer #3: No

Reviewer #4: No

Reviewer #5: No

5. Review Comments to the Author

Reviewer #1: Strengths:

1. Novel Contribution: The introduction of the Load Conservation Valley-Filling (LCVF) heuristic significantly advances electric vehicle load management.

2. Comprehensive Analysis: Strong analysis showing LCVF’s effectiveness in reducing peak charge and energy consumption across various scenarios.

3. Clear Presentation: Results are well-structured with effective tables and figures.

4. Relevance: The research is timely and relevant to the growing need for efficient energy management in smart grids.

5. Future Work: Suggestions for future research are valuable and well-articulated.

Weaknesses:

1. Statistical Analysis: More detail on statistical methods (e.g., ANOVA) is needed, including assumptions and implications.

2. Limitations Discussion: A more comprehensive discussion of LCVF's limitations, especially regarding computational complexity, is necessary.

3. Contextualization of Results: A deeper comparison with existing literature would enhance the significance of the findings.

4. Clarity and Conciseness: Some sections could be streamlined for readability and to reduce redundancy.

5. Figures and Tables: Better integration of figures and tables into the narrative would improve coherence.

Recommendations for Improvement:

1. Enhance the clarity of statistical analysis.

2. Expand on LCVF's limitations.

3. Provide a thorough literature comparison.

4. Streamline text for clarity.

5. Improve integration of figures and tables.

Reviewer #2: The manuscript presents a method for optimizing electric vehicle (EV) charging in order to reduce grid demand across many scenarios, showing notable reductions in energy consumption, improving grid stability and efficiency.

I have the following concerns about it:

1- The introduction needs to be revisited. It is lacks direction with a clear research gap.

2- The novelty of the work must be clearly addressed and discussed, compare your research with existing research findings and highlight novelty, (compare your work with existing research findings and highlight novelty).

3- There are so many works in the existing field, author should mention all new work.

4- It is recommended that the simulation results be expanded and more comparisons with state-of-the-art methods be conducted to validate the effectiveness of the proposed approach.

5- More updated references from year 2023 and 2024 can be included for literature review. Additionally, in the area of smart grid managment, You may find the following reference helpful to be added to the references list: Smart vehicle-to-grid integration strategy for enhancing distribution system performance and electric vehicle profitability, Energy, Volume 302, 2024, 131807.

Reviewer #3: Comment 1: There are several grammatical mistakes. Please work close to a native English speaker to refine the language of this manuscript.

Comment 2: The literature review section should be improved. It should be dedicated to present critical analysis of state-of-the-art related work to justify the objective of the study. Also, critical comments should be made on the results of the cited works.

Comment 3: Please carefully check recent literature and discuss/cite as you see fit and update your reference list.

Comment 4: The findings were not comprehensively discussed. It seems that the paper is a report instead of a scientific paper.

Comment 5: It would be excellent if the importance of this issue was validated by detailed research and thoroughly documented data.

Comment 6: The introduction section should be enriched by adding the following related

Reviewer #4: Comments

Title:

•Clarify the focus: specify that the heuristic is aimed at improving energy efficiency or grid management in the title for

clearer emphasis on the objective.

•Consider streamlining the title for better readability while maintaining clarity.

•Ensure the title appeals to both energy management and electric vehicle researchers by highlighting practical implications like power grid optimization.

Abstract:

•Clarify the novelty by explaining more clearly what makes LCVF heuristic innovative as compared to previous approaches like CVF and OVF.

•Consider simplifying some of the numerical results or presenting them in a more accessible way to ensure the abstract remains concise and easy to follow for a broad audience.

•Include a sentence on how these findings could be applied in real-world EV charging infrastructure or energy management systems.

•Strengthen the link between the DSM strategies and their practical implications for energy grids, offering clearer insights into how DSM contributes to the sustainability of electric grids.

Introduction:

•The introduction does a good job of outlining the challenges faced by electricity grids due to EV adoption, but it could be more concise in framing the core problem to focus the reader’s attention.

•Terms like "DSM techniques" and "Valley-Filling" are introduced well, but the explanation could be made more accessible to non-experts. Brief definitions for key terms could help general readers.

•There is some repetition, particularly around explaining the challenges of grid stress and DSM strategies. Condensing these parts could make the introduction more engaging.

•The introduction of the Load Conservation Valley-Filling (LCVF) heuristic is delayed until later in the paragraph. Consider mentioning it earlier to highlight the novelty and contribution of the research.

•The technical details (like "Peak charge (Pc)" and "load time-slots (TS)") could be briefly explained or simplified to improve clarity and accessibility.

•The English and grammar are mostly clear but could be improved for smoother flow:

Example: Instead of "in contexts where the grid operator could look at the previous patterns of EV power consumption," use "in scenarios where grid operators can analyze previous EV consumption patterns."

•Some sentences could be shortened to avoid overloading readers with complex ideas in one go.

•You can strengthen the section by discussing the broader impact of adopting LCVF, such as cost savings, energy efficiency, and real-world applicability, earlier in the introduction.

Related work section/Literature Review

•To further improve the literature review, it is essential to include a comparison table that summarizes the key findings, methodologies, and limitations of previous studies. This table will provide a clear visual reference for readers to quickly understand how the proposed approach compares to existing solutions and also mention how your studies overcome those limitations

•The literature review effectively covers various DSM techniques, but it can be better organized. Consider grouping studies more clearly by themes (e.g., EV aggregators, V2G technologies, forecasting algorithms) to enhance readability.

•There is some redundancy when describing similar techniques across different studies. Streamlining these descriptions would help reduce repetition and focus on the key contributions of each work.

•More emphasis should be placed on explaining how each referenced study specifically relates to the presented research. For instance, directly linking how previous methods fall short and how the proposed LCVF heuristic addresses those gaps would strengthen the argument.

•While the section does well in presenting technical details, Try to simplify technical jargons, some complex terms (e.g., "EVRPTW," "SoC," and "load factor") could benefit from brief explanations or simplification for readers unfamiliar with those terms.

•Transitions between studies could be smoother. Some paragraphs feel disconnected; adding linking sentences between studies could enhance the narrative flow.

•The novelty of the proposed heuristic is introduced only at the end. Consider briefly introducing the research gap and how the new approach fills it earlier in the section to keep readers engaged.

•Some studies are explained in much more detail than others. Consider providing consistent levels of detail across all referenced works or focusing more on studies that are most relevant to the proposed research.

•The English grammar is generally good, but a few sentences are overly complex. For example, “...resulted in better initial solutions and avoiding drops in local optima” could be rephrased as “...resulted in better initial solutions while avoiding local optima.” Simplifying sentence structures would improve clarity.

Methodology/ Material and Methods:

•The "Methodology" section is missing, which should clearly explain the proposed design of the new heuristic, named Load Conservation Valley-Filling (LCVF), and its detailed implications. The absence of this section makes it difficult for readers to develop a general understanding of the concept and its application.

•A flowchart or block diagram outlining the general framework of the proposed methodology is missing.

General Comments

•Consider defining key terms more clearly when they are first introduced (e.g., "valley," "peak charge," "capacity margin index") to ensure the reader fully understands their significance.

•Ensure all referenced figures (e.g., Fig. ??) are clearly labeled and include descriptive captions that summarize their relevance to the text. This will aid in comprehension.

•Provide a brief justification for why the chosen time slice (e.g., 10-minute intervals) is optimal for the VF approach in the context of electric vehicles.

•Be generous in citing the work of others. It makes the work more reliable. I have noticed at most of the places the references are missing.

Specific Comments on Algorithms

•Clarify how the parameters (e.g., upperPc, lowerPc) are initialized and adjusted in both the CVF and OVF algorithms. A flowchart or diagram illustrating the decision-making process could enhance understanding.

•Include a more detailed comparative analysis between CVF and OVF, highlighting their performance metrics (e.g., convergence speed, efficiency) based on real-world scenarios or simulations.

•When discussing the oscillatory behavior in CVF, provide quantitative data or examples to illustrate its impact on grid stability. This will underscore the importance of the OVF approach.

Comments on Load Conservation Valley-Filling (LCVF)

• Elaborate on the "state storage" concept in the LCVF algorithm. Discuss how this improves efficiency and stability compared to CVF. Providing a visual representation of how data is stored across iterations could be beneficial.

•Sensitivity Analysis: Consider incorporating a sensitivity analysis to show how variations in parameters (e.g., Pc) impact the performance of LCVF compared to CVF and OVF.

Future Directions

•Discuss potential challenges in implementing these heuristics in real-world scenarios, such as computational limits, user behavior, or grid characteristics.

•Suggest other potential applications for the VF approach beyond electric vehicles, which could demonstrate the versatility and broader relevance of your research.

Formatting

•Ensure consistent notation throughout the document, particularly for variables and parameters, to avoid confusion.

Result and discussion

•The description of the scenarios could benefit from clearer transitions between them. Consider adding a brief introduction or summary for each scenario to avoid confusion and provide a clearer flow of information.

•Cite the link of experiments and evaluation result dataset as reference

Interpretation of Results:

•While the section explains the results in terms of performance, the interpretation of why certain heuristics performed better in specific scenarios is somewhat lacking. More discussion on the reasons behind the behaviors of the CVF, OVF, and LCVF algorithms in each scenario would provide a deeper understanding.

Runtime Explanation:

•The runtime differences between the heuristics are mentioned but could use further elaboration. How does the nature of the problem complexity or dataset size contribute to the differences in runtime across scenarios?

•It might also help to quantify the runtime differences more specifically rather than just qualitatively ("1/2 the time," "1/3 the runtime"). Consider adding a table to summarize the exact times and Pc values for clearer comparison.

Visuals and Graphs:

•The discussion frequently references figures, but the figures are marked as "Fig. ??", which disrupts the flow and makes it difficult to visualize results. Correct the figure numbers and ensure they are referenced consistently.

•It would be helpful to include a discussion of how each graph supports the conclusions. For example, explicitly mention key features or patterns observed in the graphs to reinforce the written analysis.

Deeper Comparative Analysis:

•While the LCVF heuristic is often presented as the superior method, the specific trade-offs between LCVF and the other heuristics are not fully explored. Expanding on where CVF and OVF might still offer advantages (e.g., under different conditions or dataset types) would add depth to the analysis.

•The discussion on the convergence rate and resource allocation could benefit from more numerical comparisons (e.g., percentage reductions in Pc for LCVF versus CVF and OVF).

Broader Implications:

•The broader implications of the findings are not fully addressed. It would enhance the section to briefly discuss how the findings might apply to real-world EV charging systems or suggest future research directions based on the observed outcomes.

Algorithm Specifics:

•It would be valuable to include more details about the LCVF algorithm's operations, especially its ability to "conserve load" more efficiently. For readers unfamiliar with the specific techniques, a short explanation of how each algorithm allocates load in practice would enhance comprehension.

Convergence Criteria:

•The criteria for determining convergence (1% difference between iterations) are mentioned but not justified. Why was this threshold chosen, and how does it affect performance across the heuristics? A brief explanation would help clarify its importance.

Statistical Analysis

•The explanation of hypothesis testing could be streamlined. Instead of re-explaining the concept of the null hypothesis (H0) for each test, it would be beneficial to summarize these tests more concisely. This would improve readability.

•While non-parametric tests were appropriately selected for non-normally distributed data, a brief explanation of why this occurred in certain scenarios (e.g., due to skewed EV load data) could add depth to the analysis

• I found boxplots and other plots at the end but not clarified in the analysis.

•The impact of outliers should be discussed further.

•Expand the analysis to provide more practical insights into the results, particularly for practitioners who may need to decide between these heuristics based on real-world scenarios.

•While there are statistically significant differences in certain scenarios, the practical relevance of small differences, such as those observed in the flexible and increased consumption scenarios, could be further discussed to guide decision-making.

Conclusion

•While the conclusion highlights LCVF's advantages, it could benefit from a clearer explanation of the specific novel aspects that distinguish LCVF from existing methods like CVF and OVF.

•The discussion of LCVF's computational complexity and limitations is somewhat brief. Expanding on these limitations and suggesting mitigation strategies would provide a more balanced view.

•Although quantitative reductions in peak charge and energy consumption are noted, providing more context about the practical impact (e.g., cost savings, grid stability) would enhance the conclusion.

•The proposed future directions are promising, but they could be more specific. For instance, elaborating on which machine learning techniques or real-time optimization methods could be integrated would strengthen the conclusion.

Some minor points

•The list of references should be carefully checked to ensure consistency between all references and their compliance with the journal policy on referencing.

•The professional English editing is recommended. The authors should get editing help from someone with full professional proficiency in English.

•The text should be justified to aligned properly throughout the article

•The plots are presented at the end of paper are appear inappropriate at the end, it should be present at their respective position to relate those pots more concisely with your research work.

•The dataset heading is currently missing and should provide detailed information about the dataset, including its characteristics, size, and the methodology used for its utilization in the study

•Referred figure number at various position are missing which disturb the flow of article

•Include more recent references. At some places I found the references are missing

Reviewer #5: The manuscript lacks significant novelty, as the results presented do not offer new insights or advancements compared to existing literature. The study seems to replicate previous work without addressing a unique angl

6. PLOS authors have the option to publish the peer review history of their article (what does this mean?). If published, this will include your full peer review and any attached files.

Reviewer #1: **Yes: **Waqar Ahmad

Reviewer #2: No

Reviewer #3: No

Reviewer #4: No

Reviewer #5: No

---

## [Author Response · Author response to Decision Letter 0]

12 Dec 2024

REBUTTAL LETTER

Optimizing power grids: a valley-filling heuristic for energy-efficient electric vehicle charging

OUTLINE

We thank the reviewers for their detailed review of the manuscript, comments, suggestions, and criticisms. We conducted a comprehensive review of the article to apply all suggestions, correcting errors and typos, and clarifying questions pointed out by the reviewers. All revisions and additions to the manuscript are highlighted in blue. In the following section, we present the point-by-point questions raised by the reviewers and our answers.

We are extremely grateful to the reviewer for providing very constructive comments that have clearly led to the improved quality of our paper.

Journal Requirements:

-- Please ensure that your manuscript meets PLOS ONE’s style requirements, including those for file naming. The PLOS ONE style templates can be found at:

---- PLOS ONE Formatting Sample Main Body

---- PLOS ONE Formatting Sample Title Authors Affiliations

Answer:

-- The style templates provided by PLOS ONE were used to format the manuscript. We believe that the LaTeX file is fully compliant with the specified style and formatting requirements. The document was prepared using the LaTeX template provided by PLOS ONE itself, ensuring adherence to the guidelines for file naming, title structure, authors, affiliations, and the main body of the manuscript.

2) We suggest you thoroughly copyedit your manuscript for language usage, spelling, and grammar. If you do not know anyone who can help you do this, you may wish to consider employing a professional scientific editing service.

---- A copy of your manuscript showing your changes by either highlighting them or using track changes

---- A clean copy of the edited manuscript (uploaded as the new manuscript file)

Answer:

-- The manuscript has been thoroughly reviewed for language usage, spelling, and grammar. No professional editing service was employed; instead, we utilized online editing tools to enhance accuracy and clarity. The changes have been carefully incorporated, and both a highlighted version and a clean copy of the revised manuscript are provided for your review.

3) Please note that PLOS ONE has specific guidelines on code sharing for submissions in which author-generated code underpins the findings in the manuscript.

-- In these cases, we expect all author-generated code to be made available without restrictions upon publication of the work. Please review our guidelines at PLOS ONE Materials and Software Sharing and ensure that your code is shared in a way that follows best practice and facilitates reproducibility and reuse.

Answer:

-- All code used in the manuscript has been made publicly available without restrictions through the Mendeley Data platform and can be accessed at Mendeley Data - Dataset using the DOI 10.17632/vd9d6bjcm7.1. The code is also referenced within the text where the data is presented, and it can also be found in the reference list of the manuscript.

4) Thank you for stating the following financial disclosure: “Conselho Nacional de Desenvolvimento Cient́ıfico e Tecnológico Award Number: 160179/2020-3”

-- Please state what role the funders took in the study. If the funders had no role, please state: “The funders had no role in study design, data collection and analysis, decision to publish, or preparation of the manuscript.”

Answer:

-- In accordance with your instructions, the authors have removed the financial disclosure statement from the manuscript and included it in the cover letter for your review. We have ensured the disclosure fully addresses the role of the funding agency, affirming that the funders had no involvement in study design, data collection and analysis, decision to publish, or manuscript preparation.

5) Thank you for stating the following in the Acknowledgments Section of your manuscript:

-- “The authors would like to thank the Conselho Nacional de Desenvolvimento Cient́ıfico e Tecnológico (CNPq), Grant No.: 160179/2020-3, for their financial support to this work. They also extend their gratitude to the Universidade Federal de Mato Grosso do Sul (UFMS) and the Research Laboratory of High-Performance Computing Systems (LSCAD), where this work was conducted”.

Answer:

-- In accordance with the previous guidance, we have removed all funding-related and acknowledgment statements from the manuscript to ensure compliance with submission requirements. These statements, including the funding acknowledgment for the Conselho Nacional de Desenvolvimento Cient́ıfico e Tecnológico (CNPq) and our gratitude to the Universidade Federal de Mato Grosso do Sul (UFMS) and the Research Laboratory of High-Performance Computing Systems (LSCAD), have been appropriately transferred to the cover letter.

6) We note that your Data Availability Statement is currently as follows:

-- “All relevant data are within the manuscript and its Supporting Information files.”

---- The values behind the means, standard deviations, and other measures reported;

---- The values used to build graphs;

---- The points extracted from images for analysis.

Answer:

-- As per the previous requirement, we confirm that all necessary raw data and code required to replicate the study’s findings have been made available in full. This includes the values behind reported measures, graph data, and any other minimal data set components needed to reproduce the results accurately. All code used in the manuscript is publicly accessible without restrictions through the Mendeley Data platform and can be accessed via the following link: Mendeley Data - Dataset using the DOI 10.17632/vd9d6bjcm7.1.

Answers to Reviewer 1

1) Statistical Analysis: More detail on statistical methods (e.g., ANOVA) is needed, including assumptions and implications.

Answer:

-- We thank the reviewer for the feedback on our statistical data analysis. We expanded the “Statistical Analysis” section to include ANOVA assumptions verification (normality via Shapiro-Wilk test). We clarified that only the Original Scenario shows a statistically significant runtime difference, highlighting scenario-specific impacts. For borderline p-values, we noted possible use of post hoc tests for further validation (lines 848-867).

2) Limitations Discussion: A more comprehensive discussion of LCVF’s limitations, especially regarding computational complexity, is necessary.

Answer:

-- We have incorporated a detailed discussion of the LCVF heuristic’s limitations, with particular emphasis on its computational complexity. This addition addresses the scalability challenges in large-scale applications and suggests future directions for optimization, enhancing the heuristic’s feasibility in diverse grid configurations (lines 442-448;1001-1008). Thank you for highlighting this important aspect.

3) Contextualization of Results: A deeper comparison with existing literature would enhance the significance of the findings.

Answer:

-- The related work section has been improved, and a summary table was added to facilitate comparison and the significance of the findings. We would like to thank the reviewer for their insightful feedback, which helped strengthen this work.

4) Clarity and Conciseness: Some sections could be streamlined for readability and to reduce redundancy.

Answer:

-- We have performed a throughout review on all sections where redundancies were identified, rewriting the text to make it more concise. For instance, the impact of EVs on electricity grids, the description of the Valley-Filling strategy, and the contributions of the LCVF heuristic were rephrased to remove repetitive elements while retaining the essential information (lines 21-25).

5) Figures and Tables: Better integration of figures and tables into the narrative would improve coherence.

Answer:

-- Thank you for the helpful observation! We have integrated all figures and tables more smoothly into the narrative to enhance coherence and clarity.

Answers to Reviewer 2

1) The introduction needs to be revisited. It is lacks direction with a clear research gap.

Answer:

-- Thank you for your feedback. The introduction has been revised to clearly address the research gap. It now highlights the increasing EV charging demands on electrical grids and identifies the need for more adaptive and sustainable DSM strategies. The proposed Load Conservation Valley-Filling (LCVF) heuristic is introduced as an innovative solution to optimize load allocation and enhance grid stability (lines 28-33).

2) The novelty of the work must be clearly addressed and discussed, compare your research with existing research findings and highlight novelty, (compare your work with existing research findings and highlight novelty).

Answer:

-- We thank the reviewer for the observation on our paper’s novelty and comparison to existing work. The related work has been synthesized in a table to facilitate the comparison with existing research and to highlight the unique aspects of our approach. Additionally, the Discussion and Conclusion sections have been expanded to emphasize the contributions and novelties of this work to previously proposed solutions.

3) There are so many works in the existing field, author should mention all new work.

Answer:

-- Thank you for your valuable feedback. We have added recent references to enrich the discussion further and provide a comprehensive overview of the latest developments in the field (lines 3-6).

4) It is recommended that the simulation results be expanded and more comparisons with state-of-the-art methods be conducted to validate the effectiveness of the proposed approach.

Answer:

-- Thank you for the insightful suggestion. We are currently conducting additional experiments, including tests and comparisons with metaheuristic approaches, to further validate the effectiveness of the proposed method. However, due to time constraints, we were unable to add these new results in the current review.

5) More updated references from year 2023 and 2024 can be included for literature review. Additionally, in the area of smart grid management, You may find the following reference helpful to be added to the references list: Smart vehicle-to-grid integration strategy for enhancing distribution system performance and electric vehicle profitability, Energy, Volume 302, 2024, 131807.

Answer:

-- Thank you for the valuable suggestion to update more recent references from 2023 and 2024. We appreciate the recommendation to add the study "Smart vehicle-to-grid integration strategy for enhancing distribution system performance and electric vehicle profitability" (Energy, Volume 302, 2024, 131807). This reference has now been incorporated into our literature review, enhancing our discussion on smart grid management and vehicle-to-grid integration strategies (lines 92-98).

Answers to Reviewer 3

1) There are several grammatical mistakes. Please work close to a native English speaker to refine the language of this manuscript.

Answer:

-- We thank the reviewer for highlighting the grammatical errors and typos in the text. We have conducted a thorough review to address and correct these issues.

2) The literature review section should be improved. It should be dedicated to present critical analysis of state-of-the-art related work to justify the objective of the study. Also, critical comments should be made on the results of the cited works.

Answer:

-- Thank you for the insightful feedback. We have revisited the literature review section, enhancing it with a more critical analysis of state-of-the-art studies to provide a stronger justification for our study’s objectives and impacts. Additionally, we have included critical comments on the results of the cited work, offering a more comprehensive comparison and identifying gaps that our research aims to address.

3) Please carefully check recent literature and discuss/cite as you see fit and update your reference list.

Answer:

-- Thank you for the suggestion. We have thoroughly reviewed recent literature and added several updated references to enrich our discussion and ensure our study aligns with the latest developments in the field. The reference list has been updated accordingly to reflect these additions.

4) The findings were not comprehensively discussed. It seems that the paper is a report instead of a scientific paper.

Answer:

-- Thank you for your feedback. We have expanded the Discussion section to provide a more comprehensive analysis of our findings, ensuring a deeper scientific interpretation rather than a technical report. This enhancement highlights the implications, limitations, and relevance of our results within the broader context of existing research. We believe this addition strengthens the paper’s scientific contribution.

5) It would be excellent if the importance of this issue was validated by detailed research and thoroughly documented data.

Answer:

-- We’re grateful for your suggestion. We have carefully reviewed our data, and we believe they are adequately documented and replicable for any further research. We remain open to additional recommendations to ensure scientific rigor and transparency.

6) The introduction section should be enriched by adding the following related

Answer:

-- We’re thankful for your recommendation. We have improved the introduction section accordingly; however, the specific points suggested were not included in the reviewer’s feedback, so we were unable to fully address them. Nonetheless, we believe that the new introduction section now provides a clearer and more comprehensive context.

Answers to the Reviewer 4

4.1 Title

1) Clarify the focus: specify that the heuristic is aimed at improving energy efficiency or grid management in the title for clearer emphasis on the objective.

2) Consider streamlining the title for better readability while maintaining clarity.

3) Ensure the title appeals to both energy management and electric vehicle researchers by highlighting practical implications like power grid optimization.

Answer:

-- Thank you for the helpful suggestions on the title! We have applied the changes, and the new title is “Optimizing Power Grids: A Valley-Filling Heuristic for Energy-Efficient Electric Vehicle Charging”. Adding “Optimizing Power Grids” and “Energy-Efficient” clarifies the focus on optimization and energy efficiency, while simplifying to “Electric Vehicle Charging” makes it more concise. “Power Grid Optimization” will draw attention of researchers from both fields and highlight the practical implications of the work. We believe this revised title is more informative and engaging.

4.2 Abstract

1) Clarify the novelty by explaining more clearly what makes LCVF heuristic innovative as compared to previous approaches like CVF and OVF.

2) Consider simplifying some of the numerical results or presenting them in a more accessible way to ensure the abstract remains concise and easy to follow for a broad audience.

3) Include a sentence on how these findings could be applied in real-world EV charging infrastructure o

---

## [Editor Report · Decision Letter 1]

16 Dec 2024

Optimizing power grids: a valley-filling heuristic for energy-efficient electric vehicle charging

PONE-D-24-35622R1

Dear Dr. Souza,

We’re pleased to inform you that your manuscript has been judged scientifically suitable for publication and will be formally accepted for publication once it meets all outstanding technical requirements.

Kind regards,

Barry Kweh

Academic Editor

PLOS ONE

Additional Editor Comments (optional):

The authors have clarified their methodology, provided a tabulated literature review and also improved the overall grammatical structure of their manuscript.
---

## [Editor Report · Acceptance letter]

26 Dec 2024

PONE-D-24-35622R1 

PLOS ONE

Dear Dr. Souza, 

I'm pleased to inform you that your manuscript has been deemed suitable for publication in PLOS ONE. Congratulations! Your manuscript is now being handed over to our production team.

Kind regards, 

on behalf of

Dr. Barry Kweh 

Academic Editor

PLOS ONE